# MAKING, NOT TAKING, THE BEST OF N

**Ammar Khairi & Daniel D'souza & Marzieh Fadaee & Julia Kreutzer**
Cohere Labs
{ammar,juliakreutzer}@cohere.com

## ABSTRACT

Obtaining high-quality generations in modern LLMs has largely been framed as a selection problem: identifying a single *winning* generation from a diverse pool of $N$ samples, the Best-of-$N$ (BON). Yet, this approach is inherently zero-sum, discarding diverse and potentially useful information from the pool. Instead, we explore a collaborative setup, where all candidates can potentially contribute to the final *winning* generation. To this end, we propose **Fusion-of-$N$** (FUSION): a method that uses a general LLM judge to synthesize the most informative elements of each sample into a single final answer. We compare FUSION to BON in two settings, (i) **test-time scaling**, where we sample and aggregate from a single model at test-time (ii) **synthetic data generation**, where we fuse samples from a pool of diverse teachers to improve a student model. We extensively benchmark both setups across 11 languages, 3 diverse benchmarks and varying model scales. Across the bench, FUSION consistently outperforms BON showing versatility and robustness both in test-time scaling and in downstream gains from synthetic data generation. We also perform extensive analysis on FUSION, where it shows surprising strengths and robustness under challenging settings. These results show that we should shift how we think about evaluating and utilizing LLM generations from a monolithic measure of quality, to embracing their polylithic nature. This shift allows us to integrate diverse strengths, unlock latent potential, and achieve improvements that were previously inaccessible through selection alone.

## 1 INTRODUCTION

Many of today's advances in LLMs rely heavily on aggregation at inference: The dominant approach, Best-of-$N$ (BON), involves generating multiple candidates and selecting one among them as the final output. This approach has proven highly effective for test-time scaling in tasks ranging from math reasoning and translation to open-ended tasks (Snell et al., 2025; Khairi et al., 2025; Yao et al., 2023; Wang et al., 2023a), and for producing synthetic data used in fine-tuning (Jayalath et al., 2025; Muennighoff et al., 2025), especially in multilingual setups (Grattafiori et al., 2024; Dang et al., 2024; Martins et al., 2025; Hernández-Cano et al., 2025; Lai & Nissim, 2024; Hwang et al., 2025; Odumakinde et al., 2025; Rei et al., 2025). However, existing aggregation methods treat generations as competitors in a *zero-sum game*. Whether through majority voting (Brown et al., 2024), self-consistency (Wang et al., 2023a), or reward-model scoring (Ouyang et al., 2022), the goal is to find the single best answer while discarding the rest. This hard **selection** step imposes clear limitations: it discards the diversity of reasoning paths that could be combined to produce stronger answers. It wastes much of the compute spent generating samples and risks reward hacking (Skalse et al., 2022b;a; Ichihara et al., 2025): the candidate that maximizes a judge's score is not always the most correct or useful.

In today's fast-shifting LLM landscape, where leaderboard wins change hands quickly, treating quality as a single monolithic dimension is increasingly outdated. In practice, there is rarely a single "best" answer; diverse outputs often complement one another. This motivates our central question: *can we go beyond selection and design a method that makes fuller use of all generated samples?* We propose FUSION, a simple synthesis-based alternative to BON that exploits the generative abilities of LLMs to integrate complementary signals across candidates—truly making, rather than merely taking, the best of $N$.

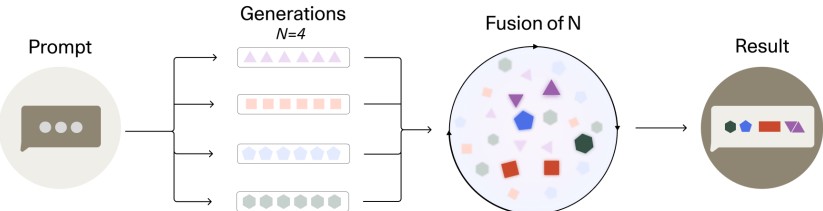

Figure 1: FUSION principle: Multiple generations (here $N = 4$, from one or multiple models) get fused into one final generation combining the strengths of each individual generation.

We treat aggregation as a **synthesis problem** rather than a selection problem. Figure 1 illustrates our idea: We use a strong LLM judge, the fusor, to integrate the complementary strengths of multiple candidates into a single answer. Our proposed Fusion-of-$N$ (FUSION) method is simple, general, versatile and can directly replace BON with no modifications beyond access to a reasonably strong generative LLM that acts as a fusor. The *polylithic* understanding of quality allows us to decompose complex problems into compositional ones that are more tractable. FUSION optimizes across samples and integrates complementary insights into a single, higher-quality answer. Going beyond the initial sample pool is especially valuable when the pool is strong and diverse, and for problems that naturally benefit from diversity. Intuitively, this mirrors how experts synthesize knowledge from multiple domains and perspectives.

We perform a comprehensive evaluation of FUSION as a replacement for BON across test-time scaling and data generation: For **test-time scaling**, we measure the effectiveness of FUSION with multiple samples from 8B and 111B models on open-ended generation and machine translation tasks. We evaluate the **impact of synthetic data** generated with FUSION in terms of data quality and downstream results after fine-tuning a 7B and a 111B model on open-ended prompts, math and factual reasoning tasks. In both setups our evaluations are spanning multiple languages to test FUSION under diverse and challenging conditions.

Our results show that synthesis is not only more effective, but also more sample-efficient: FUSION consistently outperforms BON under the same sampling budget, and in some cases even surpasses the oracle, revealing that selection is not the upper bound. It proves robust under weaker teacher pools, showing that diversity can be leveraged even when individual contributors are limited. We observe that fine-tuning on FUSION data enables models to outperform even the strongest single teacher, showing that synthesis distills collective knowledge in ways that selection cannot. Finally, our analysis provides the first detailed look into the mechanisms of synthesis, uncovering both its strengths—sample efficiency, robustness, and adaptability—and its limits on tightly constrained math tasks. To summarize, our contributions are:

**Conceptual shift from selection to synthesis.** We present compelling evidence for reframing aggregation as synthesis problem. In contrast to previous works in this direction (section 6), FUSION is simple, easily customizable and works out-of-the-box, making it an attractive substitute for BON.

**Demonstrated gains across test-time scaling and data generation.** FUSION consistently outperforms BON in both settings where candidate aggregation is used today: (i) *test-time scaling*, where it yields substantial improvements (e.g., +3.8% win-rate vs GEMINI2.5-PRO on mArenaHard-v2, +3.7 XCOMETXL on translation), and (ii) *synthetic data generation*, where it produces higher-quality datasets that drive downstream gains across diverse tasks (+2.5% on mArenaHard-V2.0 vs GEMINI2.5-FLASH, +0.8 on WMT, +1.0% on GeoFactX answer accuracy and +0.8% on reasoning quality).

**Robustness and efficiency across models and settings.** Our analysis shows that FUSION is more sample-efficient and robust than BON. It maintains high performance with smaller or weaker teacher pools, benefits from larger fusor models, and scales effectively with added test-time compute. These properties make it a practical and generalizable approach for both test-time scaling and synthetic data generation, even under constrained or imperfect conditions.

This work redefines how we measure and leverage LLM outputs. Instead of treating generations as isolated candidates, we embrace their diversity and complementary strengths, synthesizing them

into more powerful, coherent results. Our findings show that treating LLMs as collaborators and not competitors unlocks higher-quality outputs and more impact on downstream usecases, pointing toward a fundamentally more effective paradigm for large-scale language model deployment.

## 2 METHODOLOGY: FROM SELECTION TO SYNTHESIS

**Selection with Best-of-N** (BON) Given a prompt $x$, a pool of candidates $y \in Y$, and a scoring function $S$, the BON method selects the optimal candidate $y^*$ by maximizing a scalar score: $y^* = \arg\max_{y \in Y} S(y, x)$.

The scoring function could be a specialized reward model as used in rejection sampling for synthetic data generation (Grattafiori et al., 2024), or test-time scaling (Cobbe et al., 2021). The score could also be produced by a generative LLM that is prompted to predict a scalar score (Kim et al., 2024), though in practice trained classifiers often perform better, for instance many top models on Reward-Bench (Malik et al., 2025) leaderboard are sequence classifiers. These type of scoring functions are typically optimized on verifiable domains and pairwise human preferences (Cobbe et al., 2021; Ouyang et al., 2022).

**Limitations of BON** The limiting factors for selection with BON are (1) the alignment with the desired task (Lambert et al., 2020; Pan et al., 2022; Ichihara et al., 2025; Viswanathan et al., 2025), (2) and the quality of the generated sample pool (as per definition, the final generation can *only be as good as the best* of the candidates). For domains with verifiable problems, the alignment can easily be improved by scaling up training data for the reward model (Liu et al., 2025), but even with expensive ensembles (Eisenstein et al., 2024) risks of overfitting to an imperfect proxy remain (Stroebl et al., 2024). Scaling up reward model alignment is less transferrable to open domains like chat or open-ended question answering, where this signal needs to be obtained from human feedback (Huang et al., 2025; Viswanathan et al., 2025). Similarly, a poor initial sample pool can be improved by diversification (Chen et al., 2025) or optimized sampling (Khairi et al., 2025), or simply scaling up the number of samples, sometimes requiring thousands of samples for test-time scaling to be effective (Stroebl et al., 2024; Brown et al., 2024), which makes it extremely resource-intensive.

**Synthesis with Fusion-of-N** (FUSION) A fusor model $F$ (a standard LLM) generates a *new* response $y^*$ based on the input prompt $X$, and a pool of candidates $Y$: $y^* = F(x, Y), \quad y^* \notin Y$

This means that the final generation $y^*$ is conditionally dependent on the other candidates, and, can—in contrast to BON—*exceed* the original pool in quality (see Section 5). It can be seen as a form of collaborative refinement: Rather than only selecting a sample according to a monolithic notion of quality, FUSION goes beyond and productively *integrates a polylithic notion of quality into the synthesis* of a better sample. The polylithic view, meaning that we acknowledge the existence of higher and lower-quality parts in each sample, is particularly well suited for long generations for complex prompts. FUSION can "mix and match" fragments of variable size (e.g. tokens, terms, sentences, ...) that stand out in quality in each of the provided samples (see the example in fig. 15). BON is captured as a special case: the fusor still has the option to copy one whole generation if it outperforms all others for the entire sequence.

**Components of FUSION** The success of FUSION depends on the capabilities of the judge to comparatively evaluate, extract and aggregate the best parts of each generation. We will show in section 5 that there appears to be a threshold in model size that needs to be crossed for FUSION to work without any specialized training. Our analysis also shows that the choice of fusor, given a certain model size, seems less important than the composition of the sample pool. One major advantage over using a reward model, is that the FUSION prompt (ours in table 4) allows for in-context learning and adaptation *without any training*. It can be tuned to steer FUSION behavior in ambiguous cases, such as concerning safety standards (e.g. with a constitution (Bai et al., 2022)), tone or model identity, and how much it should attempt to integrate parts from all samples or also discard the worst ones. With chain-of-thought prompting (Wei et al., 2022) or reasoning models as fusors, we also have the possibility to scale up FUSION compute where desired. In preliminary experiments we found it important to instruct the model to not only focus on the best, but also discard the worst parts. We have not conducted any prompt tuning beyond that, but practitioners are invited to tune their FUSION prompt to their use cases.

# 3 EXPERIMENTAL SETUP

Our experiments span two prominent environments for BON, the first focused on **test-time scaling**, and the second focused on **synthetic data generation**. In both cases, our intervention of replacing BON by FUSION is minimal: Both methods receive the *identical set of generations* for the same prompts, but aggregate it differently to produce the final generation.

## 3.1 MODELS FOR TEST-TIME SCALING

We study the test-time scaling behavior for multilingual models of two sizes: AYA EXPANSE 8B and COMMAND A at 111B. We use temperature sampling at $T = 0.7$ to generate $N = 5$ samples from each model (see Figure 6 for various $N$). We use a competitive in-house multilingual Reward Model (RM)[1] for scoring the candidates in BON and COMMAND A as fusor in FUSION (ablation and comparison to GEMMA models (Team et al., 2025a) in fig. 5).

## 3.2 MODELS AND DATA FOR SYNTHETIC DATA GENERATION

**Models.** For synthetic data generation, we employ five open and strong models of varying size and families as teachers: GEMMA3-27B-IT, KIMI-K2-INSTRUCT, QWEN3-235B, DEEPSEEK-V3 and COMMAND A (Team et al., 2025a;b; Yang et al., 2025; DeepSeek-AI et al., 2025; Cohere et al., 2025). We sample a low temperature completion ($\tau = 0.3$) from each of them to generate the pool of samples for each prompt. From this pool, we then select one completion for supervised fine-tuning (SFT), either with RM or COMMAND A as fusor. Ablations regarding pool composition and fusor model choice will follow in table 2. For fine-tuning, we choose an 111B instruction-tuned LLM as our baseline model for our main SFT experiments, and perform an ablation with a smaller 7B Base LLM baseline (Appendix G). Finetuning hyperparameters are listed in appendix B. We do not apply test-time scaling on top of our fine-tuned models.

**General Fine-tuning Dataset.** For our main fine-tuning experiments, we randomly sample 10k prompts from UltraFeedback Binarized (UFB) (Tunstall et al., 2023), an English preference dataset with 61k pairs that was previously used to measure the impacts of data composition in fine-tuning (Odumakinde et al., 2025; Li et al., 2025b). We translate the prompts automatically into 9 languages: German, French, Spanish, Chinese, Japanese, Arabic, Korean, Italian, Portuguese.

**Reasoning Fine-tuning Dataset.** Learning to reason is often approached through synthetic data, where models imitate reasoning traces from a single teacher (Shridhar et al., 2023; Muennighoff et al., 2025; Hwang et al., 2025). Here, we apply our FUSION approach to learn to reason from multiple teachers. We add a second, smaller, batch of prompts for domain-specific reasoning tasks: We add the prompts from the GeoFactX dataset (train split) for geography-based factual reasoning, and translated s1k prompts (Hwang et al., 2025) for mathematical reasoning. The prompts are machine-translated from English and cover five and ten languages, respectively. We prompt the teachers to generate chains-of-thought and answers for training a student model (details in appendix C).

## 3.3 EVALUATION BENCHMARKS

We focus on challenging, multilingual benchmarks that test our models' *generative* abilities and cover tasks of three domains (full details in appendix D):

**Open-ended challenging prompts (Arena)** are sourced from *mArenaHard V.2* (Khairi et al., 2025) (11 languages). Quality of generations is measured in terms of win rates as determined by an LLM judge (`gpt-4o-2024-05-13`) (1) in direct comparison to the commercial GEMINI2.5-FLASH and GEMINI2.5-PRO models and (2) in head-to-head comparisons of FUSION vs BON.

**Machine Translation (WMT)** prompts are sourced from *WMT24++* (Deutsch et al., 2025; Kocmi et al., 2024a) (English to 10 languages). Quality of generations is measured with XCOMETXL (Guerreiro et al., 2024), a state-of-the-art multilingual translation evaluation metric.

---

[1]It scores an average score of 76.1 on the English RewardBench2 (Malik et al., 2025), which at the time of submission (24 Sept 2025), places it at 11th place. On multilingual RewardBench (Gureja et al., 2025) it scores an average of 87.6 across languages, beating all openly benchmarked models.

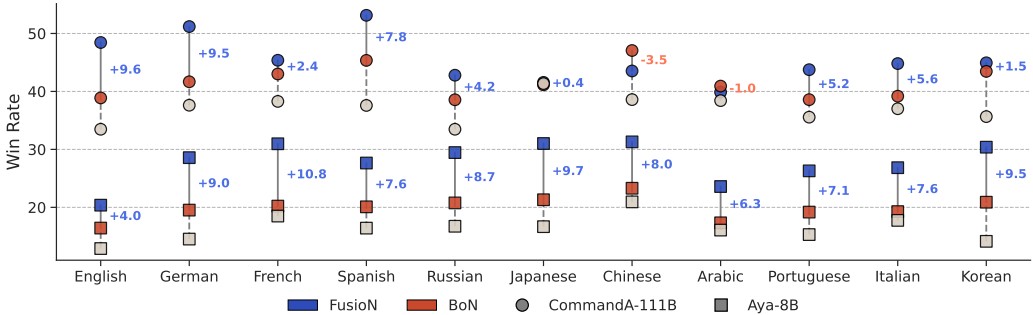

Figure 2: **Test-time scaling with** $N = 5$: FUSION raises win rates against GEMINI2.5-PRO Arena across languages. It largely outperforms BON with the same set of samples, for both AYA EXPANSE 8B and COMMAND A models. Gray markers indicate greedy baseline performance.

**Reasoning** evaluations target the reasoning fine-tuning mix and include the GeoFactX test split (Hwang et al., 2025) (5 languages) and math problems from *MGSM* (11 languages incl. English) (Shi et al., 2022). Both are evaluated in terms of accuracy of the final answers, and we additionally inspect reasoning quality for GeoFactX, following (Hwang et al., 2025).

## 4   RESULTS

### 4.1   TEST-TIME SCALING

**FUSION brings substantial improvements in multilingual open-ended generation tasks.**    We evaluate both COMMAND A and AYA EXPANSE 8B on Arena when scaling test-time compute (Section 3.1) and comparing gains from using FUSION vs BON. The results in Figure 2 show significant gains in win-rate against GEMINI2.5-PRO across both languages and models (detailed breakdown in Tables 13 and 14. For AYA EXPANSE 8B we see impressive jumps in win-rate of up to +10.8% in French. Similarly, FUSION outperforms BON for COMMAND A in 9 out of 11 languages. Surprisingly, in cases like German (+9.5%) and Spanish (+7.8%) the gains from using FUSION on only 5 samples allow COMMAND A to *win over* GEMINI2.5-PRO (absolute win-rate $> 50\%$), the top model in Arena. This special case, where fusor and sampling model are identical, FUSION can be seen as a form of very effective self-refinement (Ranaldi & Freitas, 2024). The gains from FUSION are also consistent at different scales (number of samples), tasks and in direct comparison which we investigate deeper in Section 5 and Appendix G.

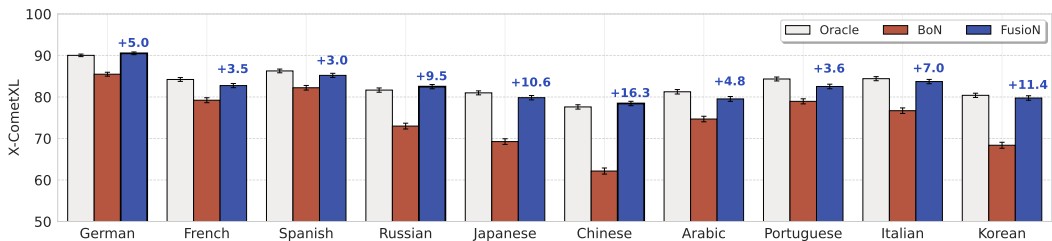

Figure 3: FUSION vs BON vs ORACLE (the highest scoring sample according to the ground truth) in Translation, error bar show std-err. Bars with bold border (German, Russian and Chinese) are cases where FUSION is outperforming the ORACLE.

**Synthesis beats selection in machine translation.**    When testing on WMT we can use the reference translation to score each candidate generation against it with the task metric XCOMETXL. We can thus identify the "oracle" among our samples, and compare its quality to the quality of samples selected by BON with its (imperfect) RM, or the sample synthesized by FUSION. Figure 3 shows the comparison for $N = 5$ generations from COMMAND A sampled at temperature $\tau = 0.7$ for the

|  |  | ar | de | en | es | fr | it | ja | ko | pt | ru | zh | *Avg* |
|---|---|---|---|---|---|---|---|---|---|---|---|---|---|
| Arena | BoN | 43.9 | 43.1 | 42.7 | 43.3 | 44.5 | 44.2 | 43.6 | 45.1 | 43.4 | 43.7 | 44.8 | *43.8* |
|  | FUSION | **45.1** | **44.3** | **48.0** | **46.2** | **48.3** | **48.4** | **43.8** | **48.4** | **45.0** | **45.2** | **46.3** | ***46.3*** |
|  | Δ | +1.2 | +1.2 | +5.3 | +2.9 | +3.8 | +4.2 | +0.2 | +3.3 | +1.6 | +1.5 | +1.5 | *+2.5* |
| WMT | BoN | 73.8 | 90.9 | - | 86.4 | 83.5 | 85.6 | 81.6 | 81.7 | 85.1 | 83.0 | 78.6 | *83.0* |
|  | FUSION | **74.6** | **91.2** | - | **87.2** | **84.3** | **86.2** | **83.1** | **82.8** | **85.5** | **83.5** | **79.8** | ***83.8*** |
|  | Δ | +0.8* | +0.3* | - | +0.8* | +0.8* | +0.6 | +1.5* | +1.1* | +0.4 | +0.5 | +1.2* | *+0.8* |

Table 1: **Downstream evaluation** of BoN/FUSION-fine-tuned 111B models on Arena (% win rate against GEMINI2.5-FLASH) and WMT (XCOMETXL, en→ ·): FUSION outperforms BoN consistently across tasks and languages. * indicates significance for WMT results according to `comet-compare` (paired t-test and bootstrap resampling (Koehn, 2004)). The baseline starts with an average score of 22.8% for Arena, and 82.0% for WMT.

WMT24++ test set. FUSION outperforms BoN with large margins across languages, reaching differences of +11.4 in Korean. More importantly, FUSION *outperforms the* ORACLE selection in the German, Russian and Chinese translation with gains of +0.8 in the latter, a meaningful improvement in terms of XCOMETXL scores. This confirms the utility of our proposed synthesis framework of aggregation. Instead of treating generations as competitors in a zero-sum game, we should treat them as collaborators whose strengths can be integrated.

## 4.2 SYNTHETIC DATA GENERATION

**FUSION yields consistent multilingual gains with downstream impact.** We compare generation and translation quality of the model fine-tuned on FUSION-generated data with the model trained on BoN-generated data in table 1 (see Appendix G for 7B results). All hyperparameters, prompts and teacher outputs are identical for both variants. Given that we only change the way we aggregate the samples, we find surprisingly notable and consistent improvements of FUSION over BoN, across languages and the two tasks. On average, the model fine-tuned on fused generations yields XCOMETXL scores +0.8 higher on WMT24++, a delta that can be expected to represent human preferences with around 73.6% accuracy, according to estimates in (Kocmi et al., 2024b).[2] Similarly, FUSION improves win-rates against GEMINI2.5-FLASH by +2.5% over BoN. With only minimal intervention in the data generation phase, the results reveal a remarkable downstream impact, underscoring the powerful ripple effect that even modest improvements in data generation can achieve. The 7B model finetuned with FUSION outperforms the one finetuned with BoN on WMT, but not Arena as we discuss in Appendix G.

**FUSION leads to better multilingual factual reasoning** Figure 4 demonstrates how the model fine-tuned on FUSION outputs outperforms the model fine-tuned on BoN in terms of answer correctness and reasoning score across four out of five languages, with a minor regression in Hindi. The fine-tuned models do not only outperform base model (by +8.1% in answer correctness on average for BoN, +9.1% for FUSION), but also the fusor model (by 3.4% and 4.4%, respectively, see full results in table 10). This validates our hypothesis that we can effectively leverage the wisdom of the crowd without being bounded by the model that performs the fusion (see also Appendix G). It is worth noting that this holds even for the languages that the fusor model (COMMAND A) officially does not support (Swahili and Thai). On MGSM, however, we found some cases where FUSION scores below BoN, which we discuss in appendix E.2.

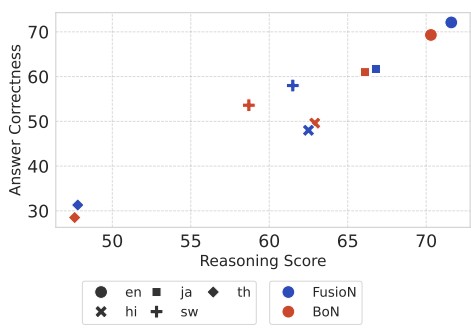

Figure 4: **Downstream evaluation on multilingual factual reasoning** on the Geo-FactX test set. FUSION outperforms BoN notably in both reasoning quality and answer correctness in 4/5 languages.

[2] https://kocmitom.github.io/MT-Thresholds/

## 5 ANALYSIS

Our results reveal consistent improvements across setups and languages with an out-of-the box fusor and a small set of samples. To find out, *where and how* FUSION is working, we conduct a range of ablations, diving deeper into specific sub-questions.

**What makes the fusor work?** In Figure 5 we approach this question from two angles: (i) the scale of the fusor (number of parameters), and (ii) how the fusor model is utilized. We evaluate the size effect by varying the fusor from the *4B* Gemma-3 to the *111B* COMMAND A measuring the resulting average win-rate of test-time scaling on Arena. We find that for FUSION (blue) **a larger scale is needed for the fusor to work out of the box**. Importantly, FUSION continuously benefits from increasing the scale of the fusor as we see an increase in win-rates of +5.5% as we go from the *27B* fusor to a *111B* fusor. When we use the same fusor models as a *rater* in BON (red) (prompt in Appendix A), smaller models fare better, but these gains vanish at scale, which aligns with the observation that even the strongest generative models such as GEMINI2.5-PRO are still outperformed by classifier RMs on classic reward scoring benchmarks (Malik et al., 2025). Overall, FUSION utilizes the judge capabilities at larger scale more effectively than BON. Smaller fusors likely need specialized training for FUSION, which related work has done for math (Qi et al., 2025; Zhao et al., 2025).

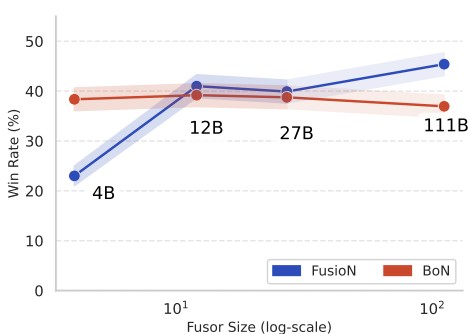

Figure 5: **Size of the fusor matters**: Small LLMs might serve well as scalar judges in BON, but generative fusion capabilities get unlocked at larger scale, here measured in win-rates on Arena, averaged across languages, shaded areas represent std-err.

**Which method is more sample-efficient?** We compare the sample efficiency of BON and FU-SION under the same test-time scaling budget. In Figure 6 we measure win-rates on Arena across four languages (See appendix G for language breakdown). We observe that FUSION is more efficient at the lower scales ($N < 10$), improving win-rate against GEMINI2.5-PRO by +6% with only $N = 2$, where BON needs twice more samples to achieve similar gains. Gains for both methods plateau beyond $N = 7$, but FUSION consistently makes fuller use of each generated sample, making **FUSION the more efficient choice for low-budget scaling**. Note that BON requires $N$ independent samples, which are parallelizable, while FUSION encodes all samples together (more details in Appendix I). Despite this, FUSION shines at small $N$, making every token count and turning even a few samples into high-quality, integrated solutions. With an efficient long-context implementation, it can achieve strong scaling performance while fully leveraging the diversity in the sample pool.

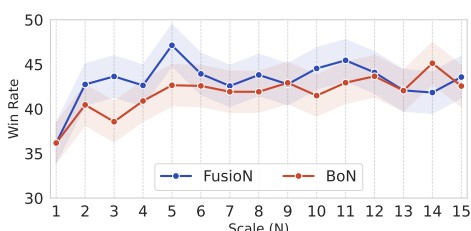

Figure 6: **Scaling test-time budget**: Win-rates are shown against GEMINI2.5-PRO on 4 languages from mArenaHard-V2.0. Shaded areas are average std-err across languages.

**How is synthetic data quality affected by the fusor and teacher pool?** The quality of the synthetic data generated is dependent on the quality of the pool of the samples and the fusor used. We measure quality of the data averaged across 10 languages using win-rates against GEMINI2.5-FLASH for 1k examples of UFB, and report in table 2 how modifications to the teacher pool and fusor affect data quality. Across all modifications we see that FUSION-generated synthetic data is of higher quality compared to the BON data with +4.4% in the default setup with all five teachers. Even when we perform FUSION with—on this benchmark—weaker DEEPSEEK-V3 (based

| # | Candidate Pool | Method | WR |
|---|---|---|---|
| 0 | CMD: 1 Sample | - | 57.9 |
| 1 | All 5 Teachers | FUSION | 65.4 |
| 2 | | BON | 61.0 |
| 3 | | Fusor=DS | 63.9 |
| 4 | Weaker Pool | FUSION | 65.0 |
| 5 | | BON | 60.9 |
| 6 | Smaller Pool | FUSION | 62.9 |
| 7 | | BON | 60.2 |
| 8 | DS: 5 Samples | FUSION | 59.0 |
| 9 | | BON | 58.9 |

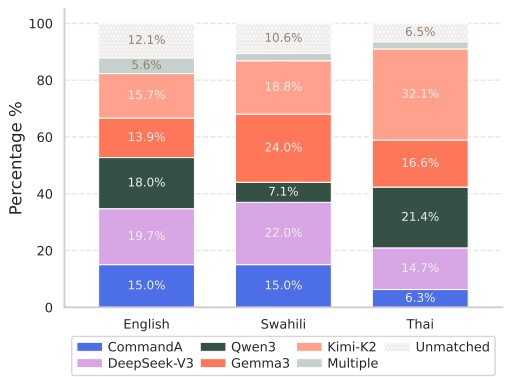

Table 2: **Ablation on pool size and diversity**: win-rates (*WR*) vs. GEMINI2.5-FLASH using 1k random UFB samples, averaged over 10 languages. *DS*: DEEPSEEK-V3. *CMD*: CommandA.

Figure 7: **Diverse teacher contributions**: Analysis of teacher contributions to the final output of FUSION on a subset of GeoFactX (50 samples per language) across unsupported languages.

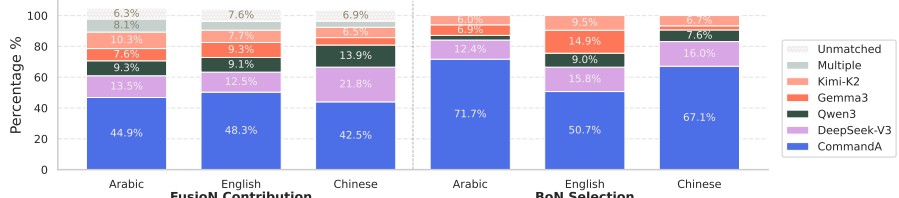

Figure 8: **Contributions to the final generation:** Analysis of different teachers in the pool contributing to the final output of FUSION or BON. For the multi-teacher we look at FUSION outputs on subset of UFB (50 samples per language) with a 5-teachers pool.

on reward scores from our internal RM) (#3), we see only a small drop in quality while still outperforming BON. When we replace GEMMA3-27B-IT with GEMMA3-4B-IT in the teacher pool (weaker teacher pool) (#4+#5), both methods are minimally affected—however, using a smaller pool of only four teachers (without KIMI-K2-INSTRUCT, #6+#7) affects FUSION proportionally more, but it still wins over BON. If we sample only from a single teacher (here DEEPSEEK-V3, #8+#9) win-rates drop substantially, highlighting the importance of diversity in the teacher pool. Overall, these ablations show that FUSION is **more robust under weaker ensembles** than BON.

**How does FUSION balance its pool?** We track the sources of contribution in FUSION by surface-level sequence matching in Figure 8(details and more bias probes in appendix F, with an example in fig. 15). We see that a large proportion of the FUSION output is directly taken from the teachers' outputs, forming a coherent synthesis. **Only a small fraction of words is *unmatched***, where the fusor adds "glue" or reformulates teacher outputs. While both methods show similar high-level preferences (favoring COMMAND A the most and GEMMA3-27B-IT the least), FUSION integrates even the less preferred ones. Finally, we inspect the contributions in the GeoFactX data, because it contains languages not officially covered by the fusor (COMMAND A). Figure 7 shows that FUSION remains robust, with its preferences shifting to utilize GEMMA3-27B-IT the most.

**Is the fusor putting a ceiling on the quality?** In Table 3, we compare the outputs of FUSION, BON, and all teachers on various metrics on the GeoFactX train set. For *answer correctness*, FUSION achieves the highest accuracy with 58.6%, despite the fusor (COMMAND A) scoring the lowest. FUSION also obtains the highest *reward score*, from our internal RM, on the GeoFactX mix—the very metric BON is optimizing. We note that FUSION has a low 98.9% *language correctness*, likely due to our fusion prompt being English-only (Appendix A), but leave studying these effects

| Model | Reward Score | Answer Correctness | Language Correctness |
|---|---|---|---|
| FUSION | **7.86** | **58.56** | 98.86 |
| BoN | 7.11 | 49.77 | **99.05** |
| CommandA | 6.30 | 40.52 | **99.61** |
| DeepSeek | 6.43 | **48.89** | 98.99 |
| Qwen-3 | 6.46 | 41.06 | 99.58 |
| Gemma-3 | 6.52 | 41.89 | 92.11 |
| Kimi-K2 | **6.64** | 46.62 | 99.56 |

Table 3: **Data analysis** for teachers and aggregation outputs on GeoFactX across languages.

to future work. These results show that **FUSION is not limited by the fusor** and is in fact more dependent on the sample pool.

**Opportunities for strengthening FUSION.** During test-time scaling we find that FUSION benefits English more than other languages. It is likely that the skills that are required to perform successful FUSION are not evenly present in all languages. Although we do find gains of FUSION over BoN in unsupported languages of the fusor for GeoFactX, BoN might be the safer choice for cross-lingual transfer to lower-resource languages (Hong et al., 2025), while generative model capabilities are still lacking behind. We also found more mixed results when testing on MGSM (appendix E.2), which might indicate that close-ended tasks are either just not well suited to be addressed by generative ensembling, or that the fusor would need specialized training for such a specialized domain that RMs are usually well trained on.

## 6 RELATED WORK

The principle of learning from ensembles has led to advances in many areas of machine learning, and can be integrated into LLMs in various forms (Wang et al., 2023b; Lee et al., 2023; Huang et al., 2024; Wan et al., 2024). In this work, we focus on *integrative output ensembling*. This approach can be seen as an instance of *Mixture-of-Agents* (MoA) (Wang et al., 2024), a framework where multiple agents organized in layers iteratively enhance the output. Our approach stands out through simplicity: We show that FUSION becomes effective already in a *single aggregation step* with a single fusor, even in diverse and challenging setups, thereby constituting an attractive alternative for BoN, which is—thanks to its simplicity—a much more widely adopted framework than MoA.

*LLM-Blender* (Jiang et al., 2023) follows a similar idea, but requires two separate modules, one for pairwise ranking, and one for fusing top-ranking outputs. It operates on the basis of pairwise comparisons (which require training a specialized model), while we pass *all outputs at once* to the fusor, so that it can evaluate them in context. Other contemporaneous related works also require training such specialized aggregator modules (Qi et al., 2025; Zhao et al., 2025; Li et al., 2025b), while our approach is effective *without any training*. These works focus primarily on reinforcement learning or reasoning for verifiable tasks like math and code. For such specific scenarios with available expert raters, Li et al. (2025a) warn that MoA might not be sufficiently robust to lower-quality inputs. For our diverse open-ended evaluations, however, we find FUSION fairly robust to the teacher pool, and sampling from a single teacher—the proposed solution by Li et al. (2025a)—performs significantly worse. Jayalath et al. (2025) find that fused single-teacher roll-outs can nevertheless provide valuable supervision in RL training, even without any fusor training.

Our approach can also be cast as *combination of parallel and sequential* test-time-scaling (Welleck et al., 2024; Snell et al., 2025), with $N$ parallel steps and one refinement step. Balancing these options can be seen as a search problem (Inoue et al., 2025). This poses an interesting avenue for future work, where FUSION operates with adaptive compute (rather than a fixed $N$+1) customized for each input. This flexibility might be needed for mimicking human cognitive processes more closely (Zhang et al., 2024). Overall, our work complements very recent advancements discovering collaborative synthesis at inference, enhancing understanding of its benefits and limitations. Even in its simplest form, our approach demonstrates gains across diverse applications, including test-time scaling and model supervision.

## 7 CONCLUSION

Our work thoroughly investigates and challenges the to-date standard practice of BON in test-time scaling and synthetic data generation. Our experiments strongly support replacing it by FUSION in these scenarios to make most of the costs that are already incurred from generating and evaluating multiple samples. Across a range of challenging multilingual tasks, FUSION consistently outperforms traditional winner-takes-all approaches like BON, delivering higher-quality outputs, greater sample efficiency, and stronger downstream performance. Importantly, FUSION leverages the strengths of multiple models, even when some are weaker, showing robustness and adaptability. These results highlight a shift in how we should think about evaluating and utilizing LLM generations: rather than measuring quality monolithically, embracing their polylithic nature allows us to integrate diverse strengths, unlock latent potential, and achieve improvements that were previously inaccessible through selection alone. FUSION points toward a more effective and sustainable paradigm for leveraging the collective capabilities of today's leading LLMs.

### ACKNOWLEDGMENTS

First, we thank our colleagues for their help in various stages of this project: Wei-Yin Ko, Kylie He and David Mora for the help with post-training, Kelly Marchisio for her advice regarding benchmarking, Thomas Euyang for the beautiful illustration, Madeline Smith for the help with communications, Sara Hooker and Ye Shen for feedback in discussions in early stages of the project, and the remaining Cohere Labs for their helpful feedback throughout all iterations of this project. Furthermore, we would like to thank Jaedong Hwang for sharing data and evaluation code for the synthetic reasoning experiments. Last but not least, we would like to thank the anonymous human and/or AI reviewers that have helped us improve the robustness of our evaluations.

### ETHICS STATEMENT

Training on synthetic data comes with inherent risks of propagating and amplifying biases (Ahn et al., 2022; Shimabucoro et al., 2024; Mohammadshahi & Ioannou, 2025). We hope that by increasing diversity in the teacher pool, we can reduce model-specific biases to propagate (as opposed to learning from one teacher only), and prevent loss of diversity in the generated data (Briesch et al., 2024).

Regardless, we cannot strictly protect the student model from adversarial teachers, probably even less so with FUSION than BON because they might be more prone to prompt injections. Our tests revealed robustness with respect to the quality of the teacher pool (section 5), but we have not tested truly adversarial inputs. We rely on the user to verify teacher suitability and potentially add any sanity checks. In contrast to BON, the FUSION framework allows for flexible instructions that could include e.g., a constitution (Bai et al., 2022) or specific safety guidelines. In practice, FUSION could also be prepended with a hard filter for unsafe or lowest-quality samples (e.g. language compliance via language identification), so that the undesired information does not even get to the aggregation stage.

We also perform additional analyses for typical LLM judge biases in appendix F, and find no evidence for self-preference, but a slight position bias, i.e. the fusor preferring samples that it is presented first more than those that come later.

We would also like to emphasize that any use of such ensembling needs to respect all terms of use and licenses of the individual teachers, which lies in the responsibility of the user.

### REPRODUCIBILITY STATEMENT

Fusor and teacher models that we use in this work are publicly available (section 3), as well as the prompts for fine-tuning. We transparently report prompts and instruction templates for LLM evaluation (appendix A), and benchmark metric implementations (section 3. Where models are not public (student model in the experiments on synthetic data generation, and reward model), we report scores on public benchmarks that allow to anchor our experiments. The data generation pipeline

that we describe in detail in appendix C is not perfectly reproducible due to inherent randomness in the sampling process. Therefore, we release synthetic data for BON and FUSION where licenses allow[3]. In addition, we follow the recommended practice for generative multilingual LLM evaluations (Kreutzer et al., 2025) and release our pairwise evaluations that rely on LLM judges[4].

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

---

Based on the provided Instruction and Generated Texts in `language`, fuse them into a better generation that combines the strength of each of them. Do so in two steps: First, compare the Generated Text with a focus on what sets them apart in terms of content, language quality and responsibility, highlighting strengths and weaknesses. Second, fuse them into a new final generation that combines the best aspects of each of them while avoiding the weaknesses.

The fused generation should be adequately responding to the instruction, sound natural to a native speaker, and be focused on conveying the most relevant and accurate information in a responsible and ethical way.

Output Format

Comparison: (short explanation of the strengths and weaknesses of each generation)
Answer: [[ The final fused generation ]]

Context

**Instruction**

`prompt`

**Generated Texts**

`generations`

Please analyse the Generated Texts, discarding any unsafe or unethical generations and provide your fused text. Remember to stick to the requested Output Format, providing first a short explanation and then putting the final fused generation inside double brackets [[]].

---

Table 4: Prompt used for FUSION, including `placeholders`. Generations are randomly shuffled and enumerated, presented one per line.

## A   PROMPTS

### A.1   FUSION

We provide the prompt used by the fusor in Table 4 . We use the same prompt across all tasks, setups, fusors and languages. Table 5 shows the prompt for using the fusor model as scalar rater. We also provide the *English Version* of the instruction prompts used in our evaluation in Table 6.

## B   FINE-TUNING HYPERPARAMETERS

We train the 111B baseline on the synthetic data generated from our UFB mix with a batch size of 16, cosine decay with peak learning rate of 5e-6 using Adam optimizer across 64 Nvidia H100 GPUs for 250 steps. For the extended mix (UFB and Math+GeoFactX) we use the same hyperparameter with increased number of steps of 323. We train the 7B models on the UFB mix with 16 GPUs with the same parameters.

## C   SYNTHETIC REASONING DATA

Hwang et al. (2025) build two datasets for improving multilingual reasoning abilities: *s1k-X* for multilingual mathematical reasoning and *GeoFactX* for geography-based multilingual factual reasoning. The multilinguality stems from automatic translation of the prompts, of the s1k dataset from (Muennighoff et al., 2025), and of synthetically created English prompts that designed to cover a variety of regions. For s1k-X the reasoning traces and answers from `Qwen-2.5-Instruct 72B` in s1k are also translated (via the Google Translate API). This has some undesired side effects where the mathematical notation or the answer formatting gets corrupted, e.g. with white spaces around LaTeX math symbols. For both datasets, we only work with the translated prompts, and use our pool of teachers to generate multilingual responses. For analysis and for evaluation, we use the human-verified answers provided in the GeoFactX dataset as ground truth for the evaluation of accurateness of answers. For s1k-X, a correctness analysis of the data is hindered by the inconsistencies in format for the (translated) silver answers of Qwen, which makes the extraction of answers non-trivial. We leave this analysis for future work.

---

Please act as a fair judge. Based on the provided Instruction and Generated Text, analyse the Generated Text and provide a 1-5 integer score. The given instruction is in `language` and the response should also be in `language` Your selection should be based on your judgment as well as the following guidelines for each possible score:

1. The Generated Text is unintelligibly written (incomplete sentences, leaps in logic, flagrant mechanical errors) or has majorly incorrect or unverifiable information.

2. The Generated Text is occasionally difficult to understand, dotted with minor factual or mechanical errors, or missing crucial formatting elements.

3. The Generated Text expresses useful information, is readable, has no factual errors, and has no more than a minor mechanical error or two. Though it may be informative to those unfamiliar with the subject matter, it is not overly insightful, engaging, or likely to hold up to expert scrutiny.

4. The Generated Text clearly expresses useful information at an expert level, is readable, and has no factual or mechanical errors. It could just use a quick adjustment with tone or length.

5. The Generated Text clearly expresses useful information at an expert level, is readable, has no factual or mechanical errors, and is the perfect length and tone with regard to the prompt.

Output Format

Analysis: xxx Answer: [[ SCORE ]] (this should be an integer from 1-5 and nothing else; the score should be enclosed in double brackets as indicated)

Evaluation Information

**Instruction**

`message`

**Generated Text**

`generation`

Please analyse the Generated Text and provide a 1-5 integer score according to the guidelines. Remember to stick to the requested Output Format, providing analysis and putting your final score (an INTEGER in 1-5) inside double brackets [[]].

---

Table 5: Prompt used for BoN with generative models, including `placeholders`

| Task | Prompt |
|------|--------|
| MGSM (en) | Solve this math problem. Give the reasoning steps before giving the final answer on the last line by itself in the format of "Answer:". Do not add anything other than the integer answer after "Answer": |
| WMT24++ | You are a professional `src_lang` to `tgt_lang` translator, tasked with providing translations suitable for use in `tgt_lang` (`tgt_country`). Your goal is to accurately convey the meaning and nuances of the original `src_lang` text while adhering to `tgt_lang` grammar, vocabulary, and cultural sensitivities. Produce only the `tgt_lang` translation, without any additional explanations or commentary. Please translate the following `src_lang` text into `tgt_lang` (`tgt_country`): `source_text` |

Table 6: Instruction prompts used for evaluation, including task-specific `placeholders`. MGSM prompts are taken from the simple-evals library, we only list the English one here but use them in the respective target languages.

For evaluation of fine-tuned models on the test split on GeoFactX, we follow the procedure in (Hwang et al., 2025). We prompt a LLM judge (here GPT-4O, deviating from (Hwang et al., 2025) which uses Qwen, as we wanted avoid self-bias) to score the reasoning traces for quality. We compare the final answer in the generation against the list of the correct answers provided in the task following the implementation by Hwang et al. (2025), and also verify the language of the response according to their implementation.[5]

## D  EVALUATION

We describe our set of evaluation benchmarks in more detail.

**mArenaHard V.2** (short: *Arena*):[6] This data contains 498 translated challenging prompts from mArenaHard-V2.0[7] across 23 languages (Khairi et al., 2025). Quality of generations is measured in terms of win rates in direct comparison to the commercial GEMINI2.5-FLASHand GEMINI2.5-PROmodels in addition to head-to-head comparison of FUSION vs BON. We mainly compare against GEMINI2.5-PRO in the test-time scaling environment where we use production ready LLMs with extra compute. In the synthetic data generation environment, we benchmark weaker LLMs fine-tuned on a small synthetic dataset, hence we switch to GEMINI2.5-FLASH. The pairwise comparison is as done by a LLM judge, here GPT-4O. We focus on a subset of 11 languages: English (en), German (de), French (fr), Spanish (es), Russian (ru), Japanese (ja), Chinese (zh), Arabic (ar), Korean (ko), Portuguese (pt), and Italian (it).

**WMT24++** (short: *WMT*):[8] This dataset contains translation problems sourced from the WMT 2024 machine translation shared task (Kocmi et al., 2024a) expanded to more languages (Deutsch et al., 2025). Quality of generations is measured with XCOMET-XL,[9] a state-of-the-art multilingual translation evaluation metric (Guerreiro et al., 2024). We use the prompt in Appendix A and we focus on translating from English to the following languages: Arabic (ar), German (de), Spanish (es), French (fr), Italian (it), Japanese (ja), Korean (ko), Portuguese (pt), Russian (ru), Chinese (zh).

**MGSM**: This benchmark contains 250 mathematical problems at grade-school level in 11 languages (bn, de, en, es, fr, ja, ru, sw, te, th, zh), originally translated from English (Shi et al., 2022). We prompt models to think step by step before outputting the final answer, following the `simple-evals` implementation,[10]. The evaluation metric is the accuracy of the final answer.

**GeoFactX** We follow the prompting and evaluation process recommended by Hwang et al. (2025) and evaluate reasoning traces and final answers with an LLM judge and against gold answers, respectively (details in appendix C).

## E  ADDITIONAL RESULTS

### E.1  TEST-TIME SCALING FOR MATH

We include additional analysis of FUSION on mathematical reasoning. On the MGSM benchmark (250 prompts per language), FUSION performs on par with BON, with differences within the standard error across languages (fig. 9). To provide a more statistically reliable comparison, we evaluate on the larger MMATH benchmark (Luo et al., 2025), which contains 374 examples per language across 10 languages. On MMATH, FUSION achieves 85.1% accuracy averaged across 10 languages, outperforming BON at 84.0% (+1.1%, ). Per-language results (fig. 10) show that FUSION matches or exceeds BON in 9 out of 10 languages. The *Oracle* upper bound of 90.6% indicates room for improvement for both methods on this benchmark.

---

[5]https://github.com/jd730/M2A/tree/main

[6]https://huggingface.co/datasets/CohereLabs/m-ArenaHard-v2.0

[7]https://github.com/lmarena/arena-hard-auto/tree/main/data/arena-hard-v2.0

[8]https://huggingface.co/datasets/google/wmt24pp

[9]https://huggingface.co/Unbabel/XCOMET-XL

[10]https://github.com/openai/simple-evals/tree/main

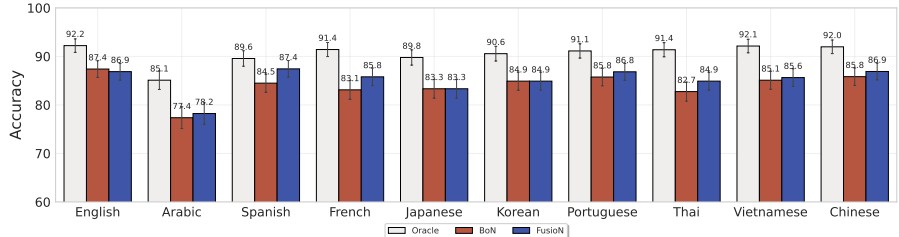

Figure 9: **Test-time Performance on MGSM**. We find that BON has a slight but not strictly significant advantage for 5/6 languages (within standard error). The performance of both methods is close to the *Oracle*.

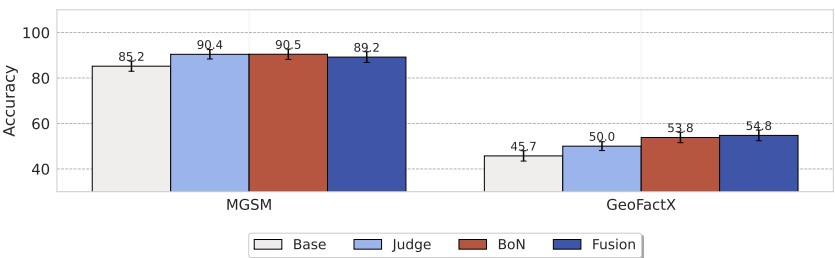

Figure 10: **Test-time Performance on MMATH.** Accuracy of BON, FUSION, and the *Oracle* across 10 languages ($N$=5, $T$=0.7) with Command A and four teacher models (∼374 examples per language). Error bars show standard error. FUSION matches or exceeds BON in 9 out of 10 languages.

We further investigate two targeted modifications for math reasoning. First, we test an adaptive reward-score filter that removes candidates scoring below a 5% threshold relative to the pool maximum before passing them to the fusor. This pre-filtering yields minor improvements in 5 out of 7 languages on MGSM, as detailed in table 7. Second, we ablate the fusion prompt design on MMATH, testing both a math-specific prompt (e.g., emphasizing reasoning correctness) and translating the prompt into the target language. Neither modification yields a statistically significant change, with all variants within ±0.1% of the baseline 72.8% accuracy, see table 8.

Taken together, these results confirm that FUSION works out of the box for mathematical reasoning without requiring task-specific prompt tuning. The magnitude of gains over BON is smaller than observed in translation, open-ended generation and world knowledge reasoning tasks, where output quality is more compositional.

## E.2 SYNTHETIC DATA GENERATION

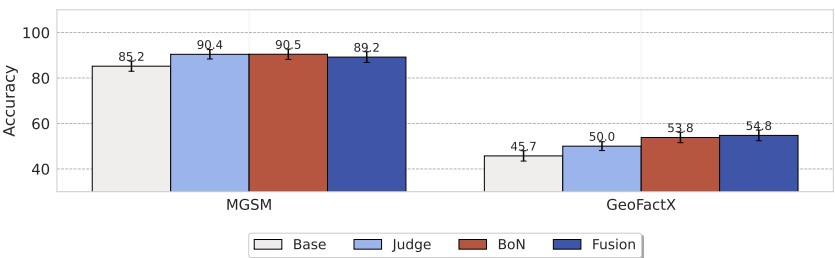

Figure 11: **Comparison of downstream performance on close-ended tasks MGSM and Geo-FactX**. We find while on math BON has the best performance, FUSION has higher accuracy on GeoFactX, outperforming the fusion judge as well. Error bars show std-err averaged across languages.

**Reasoning tasks** Figure 11 compares the performance on the two reasoning tasks for the models trained on FUSION vs BON in relation to the performance of the JUDGE model, i.e., the fusor

| Language | FUSION | FUSION + Adaptive | Δ |
|----------|--------|-------------------|------|
| fr | 85.2 | **86.8** | +1.6 |
| es | 90.8 | **92.0** | +1.2 |
| ru | 94.0 | **94.4** | +0.4 |
| ja | 90.0 | **90.4** | +0.4 |
| zh | 90.0 | **90.4** | +0.4 |

Table 7: MGSM accuracy (%) for Command A with and without adaptive reward-score filtering (5% threshold) on languages where the filter has an effect ($N$=5).

| Method | Accuracy | Std. Err. |
|--------|----------|-----------|
| FusioN | 72.8 | 2.3 |
| FusioN + Math Prompt | **72.9** | 2.3 |
| FusioN + Math Prompt + Target Language | 72.8 | 2.3 |

Table 8: Fusion prompt ablation on MMATH: test-time scaling accuracy (%) with Command A across 6 languages ($N$=5). Neither math-specific nor localized prompts yield significant improvements.

model (COMMAND A) and also the BASE model that we start fine-tuning from. For MGSM, we find that the FUSION accuracy lags behind both BON -1.2% and the JUDGE performance of -1.5%, indicating that while FUSION is overall beneficial for improving downstream math accuracy (+3.6% above BASE), it is not the optimal choice in this case (with a sligthly more pronounced gap than for the test-time scaling experiment in appendix E.1). But the facts that (1) BON does not improve the performance of the fine-tuned model beyond the fusor's performance, and that (2) the baseline model performs already surprisingly strong, make us wonder whether these results could also be due to the interplay between data seen in prior training steps, in fine-tuning, and also in the fusor model. Since the s1k dataset is quite popular, it might have been part of training (in English) of the fusor and the baseline already. For the factual QA domain we see in stark contrast, that with clearly unseen data, FUSION effects stand out more. The model fine-tuned on FUSION achieves the best accuracy with +1.0% gains over BON and an impressive +4.8% compared to fusor JUDGE.

## F    CONTRIBUTION ANALYSIS

**Measuring contributions**    We inspect FUSION outputs and compare them with the teacher outputs with string matching. While this does not capture semantic rephrasings, it does give us an idea how much the fusor can directly copy and paste blocks of the teacher outputs. Parts of the FUSION output that we cannot directly find in any teacher's output, we mark as "unmatched"—this is where we might have some close semantic matches or also just some "glue" work to connect parts from different teachers. The matching procedure works as follows:

1. Finds all matching blocks between the fused string and each teacher string. We make use of the `difflib` library[11] and use their `SequenceMatcher` to detect the longest contiguous matching subsequences.

2. Resolve attribution for each character: Retrieve matching blocks that cover it, assume the teacher with the longest match wins. If there is a tie: mark it as "multiple". If there is no match, mark it as "unmatched".

3. Calculate contribution statistics for each teacher: Count how many characters of the fused generation it was attributed to.

**Disentangling fusor and teacher pool**    The resulting contribution statistics can be compared with BON selections that chooses one teacher for each sequence. In Figure 12 we do this on a small subset of the UFB data mix covering languages Arabic, English and Chinese. We look at pool of teachers

---

[11] https://docs.python.org/3/library/difflib.html

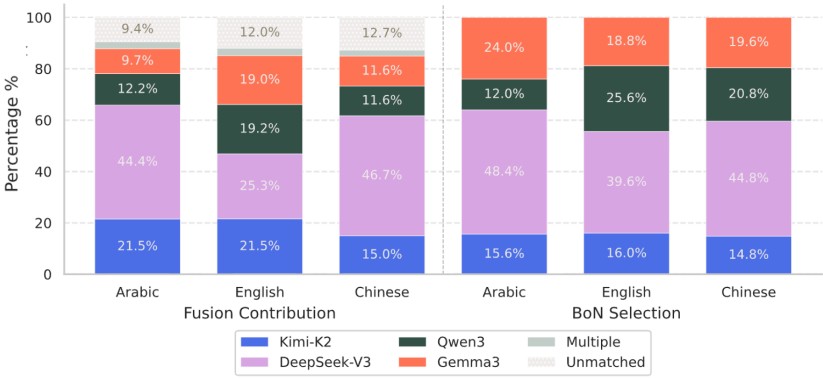

Figure 12: **FUSION contributions without fusor in the teacher pool:** Analysis of different teachers in the pool contribution to the final output of FUSION when the fusor model is *not* in the pool. We look at FUSION outputs on subset of UFB (50 samples per language)

that does not include the fusor (COMMAND A) to study the effect of the fusor self-bias. Similar to what we found in Figure 8 where FUSION and BON had their highest contribution from COMMAND A (the teacher that we now removed), in Figure 12 the methods also have same preference, agreeing on DEEPSEEK-V3 as their favorite (previously second-ranking when COMMAND A was in the pool). This consistent preference lets us conclude that FUSION does not suffer from self-bias with the fusor able to reliably find the best samples in the pool, whether or not the fusor samples is one of them.

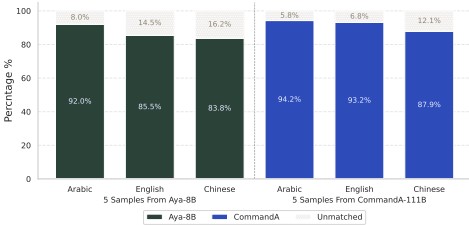

(a) Across AYA EXPANSE 8B and COMMAND A

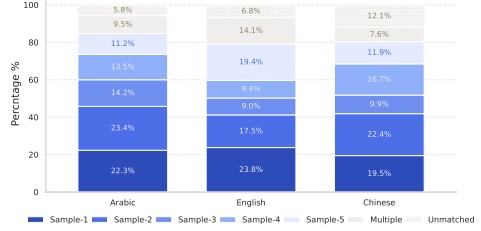

(b) Across samples from COMMAND A

Figure 13: **FUSION contributions in test-time scaling:** Analysis of how different samples in test-time scaling contribute to the final output of FUSION: (a) with different candidate models (b) based on samples order. We look at FUSION outputs on subset of mArenaHard-v2. (50 samples per language)

**Contributions in test-time scaling** In Figure 13 we perform the contribution analysis on the test-time scaling setup, where the samples in the pool are coming from a single model. First, we examine the effect of changing this single model on the FUSION preference. We are mostly interested in the case when the fusor is much larger than the candidate model as one would assume the fusor may opt to replace all of the weaker model outputs with its own preference in the fusion which would result in higher *unmatched* rate. However, in Figure 13a we see that for both the small candidate model AYA EXPANSE 8B and the larger one COMMAND A the fusor has a small *unmatched* rate. Albeit larger for AYA EXPANSE 8B, it demonstrate the fusor outputs are almost always more than 80% from the content in the samples.

**Position bias** We consider another type of possible bias in Figure 13b, where we visualize the contribution analysis based on the order of the samples in the FUSION prompt. The samples are always shuffled before being formatted (see Table 4) and we analyze the order based on what the order the fusor sees. We find while not large in magnitude, across the three languages, the fusor outputs have the highest contribution rations in the first and second samples.

Can a comedian's use of audience interaction enhance their delivery and stage presence?

==========Qwen3=============
Yes, a comedian's use of **audience** interaction can significantly enhance both their **delivery** and **stage** presence. When used effectively, audience interaction can transform a standard stand-up performance into a more dynamic, engaging, and memorable experience. Here's how:

—

### **Enhances** Delivery

1. **Creates** Natural Rhythm and Pacing
Interacting with the audience can break up the monotony of a scripted set, allowing for natural pauses, laughs, and transitions. This variation in pacing can make the performance feel more spontaneous and alive.

2. **Provides** Real-Time Feedback
Audience reactions during interactions give the comedian immediate feedback, helping...
==========Gemma3=============
Absolutely! A comedian's use of audience interaction can *significantly* enhance their delivery and stage presence. It's a powerful tool, and when done well, it's a huge part of what separates a good comedian from a *great* one. Here's a breakdown of how it works, and the benefits:

**How** Audience Interaction Enhances Delivery & Stage Presence:

* **Breaks** the Fourth Wall: It immediately makes the performance less of a one-way lecture and more of a shared experience. This creates a more intimate and engaging atmosphere.
  **Improves** Timing & Reactivity: Good audience work forces a comedian to be present* and react in the moment. This sharpens their comedic timing, as they have to...
==========DeepSeek-V3=============
Absolutely! A comedian's use of audience interaction can significantly enhance their delivery and stage presence in several ways:

### 1. **Engagement** & Connection
- Direct interaction makes the audience feel involved, creating a personal connection that boosts engagement.
- It breaks the "fourth wall," making the performance feel more spontaneous and alive.

### 2. **Adaptability** & Improvisation
- Skilled crowd work demonstrates quick wit, making the material feel fresh and unique to each show.
- It allows comedians to riff off real-time reactions, adding unpredictability and excitement.

### 3. **Energy** & Authenticity
- A dynamic back-and-forth keeps the energy high...
==========CommandA=============
Absolutely, a comedian's use of audience interaction can significantly enhance their delivery and stage presence in several ways:

1. **Engagement** and Connection: Interacting with the audience creates a sense of inclusion and connection. When a comedian involves the audience directly, it breaks the fourth wall and makes the performance feel more personal and intimate. This engagement can lead to a more receptive and responsive crowd, which in turn boosts the comedian's confidence and energy on stage.

**Example**: Asking for volunteers or incorporating audience members into bits can create memorable moments that resonate with the entire crowd.

2. **Improvisation** and Adaptability: Au...

Figure 14: An excerpt of the outputs of teachers to the FUSION example show in Figure 15

> **Can a comedian's use of audience interaction enhance their delivery and stage presence?**
>
> Yes, a comedian's use of **audience** interaction can significantly enhance both their **delivery** and **stage** presence. When executed skillfully, it transforms a standard stand-up performance into a dynamic, engaging, and personalized experience. Here's how:
>
> ### **Enhancing** Delivery
> 1. **Improves** Timing & Reactivity: Audience interaction forces comedians to be present and adapt in real-time, sharpening their comedic timing and improvisation skills.
> 2. **Creates** Natural Rhythm: It breaks up scripted material, adding spontaneity and keeping the performance lively.
> 3. **Allows** for Callbacks & Customization: Referencing earlier interactions or tailoring jokes to the crowd's responses makes the material feel fresh and relevant.
>
> ### **Boosting** Stage Presence
> 1. **Builds** Connection & Trust: Breaking the "fourth wall" fosters intimacy and makes the audience feel valued, strengthening the comedian's presence.
> 2. **Showcases** Confidence & Control: Handling unpredictable responses with humor demonstrates quick thinking and professionalism.
> 3. **Humanizes** the Performer: Sharing moments with the audience makes the comedian more relatable and approachable.
>
> ### **Types** of Audience Interaction
> - **Call** & Response: Builds energy and involvement.
> - **Heckle** Responses: Risky but rewarding when handled with wit.
> - **Playful** Banter: Engages specific audience members while maintaining sensitivity.
> - **Improvised** Bits: High-level crowd work that showcases exceptional skills.
>
> ### **Potential** Pitfalls
> - **Bombing**: Interactions can fall flat, killing momentum.
> - **Offending** Someone: Missteps can alienate audience members.
> - **Losing** Control: Disruptive audience members can derail the performance.
> - **Forcing** It: Unnatural interactions feel awkward and contrived.
>
> ### **Great** Examples
> - **Dave** Chappelle: Masters crowd work with charm and wit.
> - **Jerry** Seinfeld: Uses subtle, observational interactions seamlessly.
> - **Hannah** Gadsby: Deepens emotional impact through audience engagement.
>
> ### **Conclusion**
> When done well, audience interaction is a vital tool that elevates a comedian's delivery, strengthens their stage presence, and creates a memorable, shared experience. It requires practice, sensitivity, and a good read of the room, but the rewards are well worth the effort.

Figure 15: Contribution analysis for a sample FUSION ouput with text colored based on the respective source: CommandA, DeepSeek-V3, Qwen3, Gemma3, multiple, and unmatched. Individual teacher generations are in Figure 14.

Finally in Figure 14 and we provide an example prompt with excerpts from teachers outputs, and in Figure 15 we show the full FUSION output color coded according to our contribution analysis.

## G  DOWNSTREAM RESULTS FOR A SMALLER MODEL

To evaluate the effectiveness of our method and generated data across different scales, we applied our synthetic data generation and SFT pipeline to a smaller baseline. We followed the same setup as used for the 111B model. The 7B baseline is a base model that have not undergone any post-training

| Language | mArenaHard-V2.0 (win-rate, in %) | | | WMT24++ (XCometXL; en→ ·) | | |
|---|---|---|---|---|---|---|
| (+region for WMT) | BoN | FUSION | Δ | BoN | FUSION | Δ |
| ar (SA) | 18.3 | 15.6 | -2.7 | 65.4 | 66.3 | +0.9 |
| de (DE) | 16.8 | 16.8 | 0.0 | 86.8 | 87.1 | +0.3 |
| en | 14.9 | 14.1 | -0.8 | - | - | - |
| es (MX) | 19.6 | 17.6 | -2.0 | 81.0 | 81.5 | +0.5 |
| fr (FR) | 22.4 | 17.2 | -5.2 | 77.4 | 77.5 | +0.1 |
| it (IT) | 19.4 | 17.2 | -2.2 | 80.1 | 80.4 | +0.3 |
| ja (JP) | 17.0 | 19.5 | +2.5 | 70.8 | 72.1 | +1.3 |
| ko (KR) | 17.9 | 14.0 | -3.9 | 72.0 | 72.5 | +0.5 |
| pt (PT) | 16.9 | 16.6 | -0.3 | 79.8 | 80.2 | +0.4 |
| ru (RU) | 12.8 | 19.0 | +6.2 | 74.8 | 74.5 | -0.3 |
| zh (CN) | 19.8 | 20.9 | +1.1 | 70.9 | 71.6 | +0.7 |
| *Avg* | *17.8* | *17.1* | *-0.7* | *75.6* | *76.4* | *+0.5* |

Table 9: **Downstream evaluation results on 7B Models** of BoN/FUSION-fine-tuned 7B models on mArenaHard-V2.0 (win rate against GEMINI2.5-PRO as judged by GPT-4O) and WMT24++ (XCOMETXL).

| | Reasoning Score (LLM score, in %) | | | | | Answer Correctness (Accuracy, in %) | | | | |
|---|---|---|---|---|---|---|---|---|---|---|
| | BoN | FUSION | Δ | *Baseline* | *Fusor* | BoN | FUSION | Δ | *Baseline* | *Fusor* |
| en | 70.3 | 71.6 | +1.3 | 66.5 | 69.2 | 69.3 | 72.1 | +2.8 | 75.2 | 73.2 |
| hi | 62.9 | 62.5 | -0.4 | 54.3 | 61.6 | 49.6 | 48.0 | -1.6 | 45.1 | 51.6 |
| ja | 66.1 | 66.8 | +0.7 | 56.7 | 61.6 | 61.0 | 61.7 | +0.7 | 57.5 | 59.3 |
| sw | 58.7 | 61.5 | +2.8 | 31.7 | 44.8 | 53.6 | 58.0 | +4.4 | 35.3 | 44.0 |
| th | 47.6 | 47.8 | +0.2 | 34.2 | 39.4 | 28.5 | 31.3 | +2.8 | 22.8 | 25.6 |
| *Avg* | *61.1* | *62.2* | *+0.9* | *48.7* | *55.3* | *52.4* | *54.2* | *+1.8* | *47.2* | *50.7* |

Table 10: **Downstream evaluation on multilingual factual reasoning**, as measured on the Geo-FactX test set. FUSION outperforms BoN both in terms of reasoning quality and answer correctness, with the exception of Hindi. *Baseline* is model we used to finetune and *Fusor* is the model used for fusing the generations.

stages. As shown in Table 9, the downstream results vary. While FUSION remains more effective than BoN in WMT24, improving performance in every language, we observed no gains significant over BoN in mArenaHard-V2.0. This suggests that the smaller model requires more parameter tuning to achieve an optimal setup for SFT to be effective in downstream performance, especially the relatively small size of out synthetic dataset.

We also report the detailed breakdown of results on downstream gains in the multilingual factual reasoning benchmark from the finetuned 111B model in table 10. The *Fusor* and *Baseline* help anchor the gain and provide context about the magnitude of the finetuning gains in general. We find that fine-tuning consistently improve performance over the baseline—with one exception in Hindi. More importantly FUSION also outperform the *Fusor* in most results.

In Figure 16, we move from comparing against GEMINI2.5-PRO as our reference and directly evaluate FUSION against BoN in pairwise head-to-head setup judge by GPT-4O. We use AYA EXPANSE 8B and COMMAND A to generate 5 samples on Arena, and aggregate with either one of our methods. We find significant gain that resemble what we observe in Figure 2: FUSION outperform BoN across languages with large gains in magnitude for AYA EXPANSE 8B and lower but also impressive margins in for COMMAND A (up to +55.2% win-rate in Italian).

We perform a similar comparison in Figure 17 between the finetuned models at the 111B and 7B. We see varying results across languages in direct head-to-head comparison between the FUSION finetuned and BoN finetuned 111B models. For the 7B we see that BoN scores better across all languages.

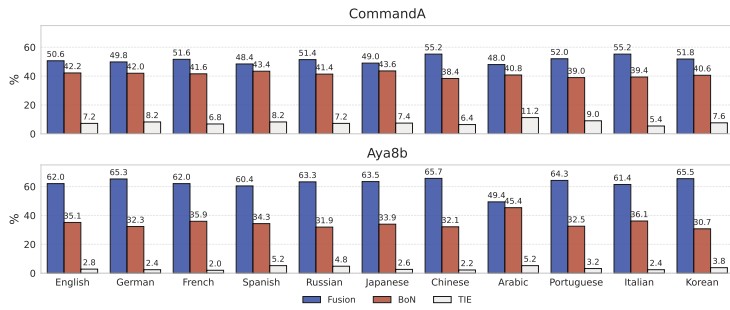

Figure 16: **Test Time Scaling** Head-to-Head Comparison of FUSION vs BON on mArenaHard-V2, Judged by GPT-4O. We consistently see that FUSION results in better final sample compared to FUSION for both AYA EXPANSE 8B and COMMAND A.

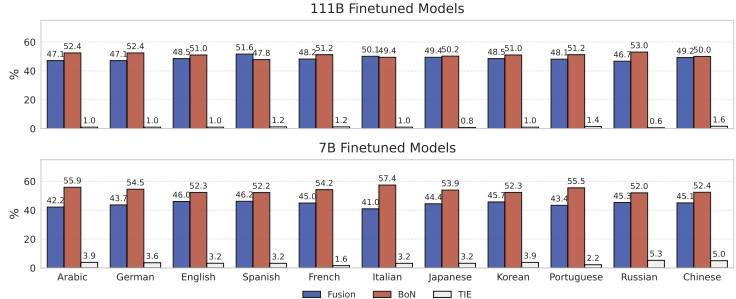

Figure 17: **Synthetic Data Finetuning** Head-to-Head Comparison of og the models finetuned with FUSION vs BON on mArenaHard-V2, Judged by GPT-4O. We see varying results across languages in the 111B case, while BON is better on the 7B level.

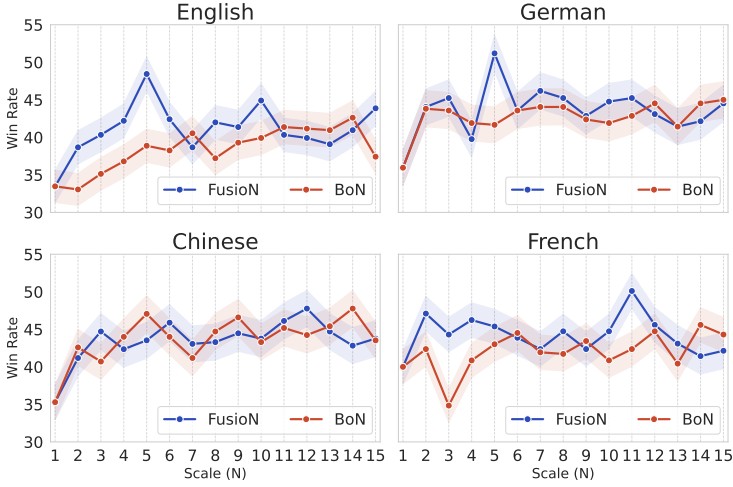

Figure 18: Head-to-Head Comparison of FUSION vs BON on mArenaHard-V2, Judged by GPT-4O. We consistently see that FUSION results in better final sample compared to FUSION for both AYA EXPANSE 8B and COMMAND A.

In Figure 18 we provide a breakdown on the scaling plots (win-rates on mArenaHard-V2.0 vs GEMINI2.5-PRO) across languages. We can see that in all subplots that FUSION grows faster than BON, with magnitudes depending on the language.

FUSION's synthesis over selection paradigm makes it more synergistic with improved sampling methods, as it leverages higher-quality samples more effectively than BON. In Table 11, we compare both methods in the test-time scaling setup by measuring the win-rate of $N=5$ samples against a single greedy sample across 7 languages. The 5 samples are generated at temperature 0.7 with Hedged Sampling (Khairi et al., 2025) and min-p sampling (Minh et al., 2024). The results show that FUSION benefits substantially more from improved sampling than BON, achieving more than twice higher win-rate gains (+33.0 vs. +12.5 for AYA EXPANSE 8B, +16.6 vs. +5.0 for COMMAND A).

| Model | BON | FUSION |
|---|---|---|
| Aya-Expanse 8B | +12.5 | **+33.0** |
| Command A | +5.0 | **+16.6** |

Table 11: Win-rate improvement over greedy (%) with Hedged Sampling and min-p on mArenaHard-V2, averaged across 7 languages with $N=5$.

We conduct additional experiments on WMT to examine why FUSION can surpass not only BON but also the oracle by design (Section 4). We compute BLEU as a complementary MT metric (as implemented in `sacrebleu` (Post, 2018), with tokenizers configured per target language) for the test-time scaling experiments with COMMAND A reported in Figure 3. Table 12 shows that FUSION outperforms the *oracle* in 4 out of 9 languages in aggregate (de, ja, ko, zh). The rightmost column reports the percentage of test sentences where FUSION surpasses the oracle, which ranges from 12% (es, fr) to 36% (ko), demonstrating that FUSION can improve upon the best individual sample in a non-trivial fraction of cases across all languages.

| Language | *Oracle* | BON | FUSION |
|---|---|---|---|
| de | 29.39 | 24.98 | **29.77** (26%) |
| es | 41.41 | 36.65 | 39.06 (12%) |
| fr | 38.00 | 33.00 | 35.63 (12%) |
| it | 34.86 | 28.43 | 34.20 (17%) |
| ja | 22.61 | 17.35 | **22.80** (34%) |
| ko | 24.61 | 18.54 | **25.41** (36%) |
| pt | 27.94 | 24.01 | 26.92 (14%) |
| ru | 21.13 | 16.96 | 21.09 (28%) |
| zh | 36.92 | 26.59 | **37.68** (34%) |

Table 12: BLEU scores (sacrebleu) for COMMAND A test-time scaling on WMT. *Oracle* selects the sample with the highest sentence-level BLEU. Percentages indicate the fraction of sentences where FUSION surpasses the oracle. **Bold** marks cases where FUSION exceeds the oracle in aggregate.

## H ENSURING ROBUSTNESS IN EVALUATIONS

In Tables 13 and 14, we provide a detailed breakdown of the test-time scaling win-rate results from Section 4. The tables report the number of samples evaluated in each language, the win rates of FUSION and BON, as well as the corresponding confidence intervals (CIs) and standard errors, computed according to the method proposed by Miller (2024).

Although our internal RM is a strong baseline to use for BON, we provided additional comparison of FUSION vs BON using an open-source reward model for better transparency and reproducibility of our work. To this end, we use SKYWORK-REWARD-V2-LLAMA-3.1-8B the currently leading open-source reward model on *RewardBench2* at the time of writing. In table Table 15, we report test-time scaling win-rate results (N=5) for both the internal reward model and the open Skywork-V2 on mArenaHard-V2, averaged across 11 languages. These results confirm FUSION's superiority compared to BON under SOTA reward models and enhance reproducibility.

We use GPT-4o as the judge in our LLM-as-judge evaluations, as it has been shown to have the highest correlation with human judges across languages in recent work Kocmi et al. (2025). However, relying on a single LLM judge still carries a risk of bias that might affect our results. To address this concern, we repeat our test-time scaling evaluation on mArenaHard-V2.0 with Claude Opus 4.1 as a second judge for a subset of languages. Table 16 shows the win-rate differences between FUSION and BON in the test-time scaling setup where we compare against GEMINI2.5-PRO. The signal is consistent across both judges: FUSION outperforms BON in three out of four languages under both judges, with varying magnitudes.

| Lang (# pairs) | Method | Win Rate | 95% CI | Std. Err. |
|---|---|---|---|---|
| ar (479) | BoN | 0.17 | (-0.56, -0.70) | 0.03 |
| | FusioN | 0.24 | (-0.44, -0.59) | 0.04 |
| de (420) | BoN | 0.20 | (-0.52, -0.68) | 0.04 |
| | FusioN | 0.29 | (-0.33, -0.50) | 0.04 |
| en (481) | BoN | 0.16 | (-0.60, -0.74) | 0.03 |
| | FusioN | 0.20 | (-0.51, -0.66) | 0.04 |
| es (463) | BoN | 0.20 | (-0.51, -0.66) | 0.04 |
| | FusioN | 0.28 | (-0.35, -0.52) | 0.04 |
| fr (465) | BoN | 0.20 | (-0.51, -0.66) | 0.04 |
| | FusioN | 0.31 | (-0.29, -0.45) | 0.04 |
| it (462) | BoN | 0.19 | (-0.54, -0.68) | 0.04 |
| | FusioN | 0.27 | (-0.37, -0.53) | 0.04 |
| ja (474) | BoN | 0.21 | (-0.49, -0.64) | 0.04 |
| | FusioN | 0.31 | (-0.29, -0.46) | 0.04 |
| ko (474) | BoN | 0.21 | (-0.49, -0.64) | 0.04 |
| | FusioN | 0.30 | (-0.31, -0.47) | 0.04 |
| pt (464) | BoN | 0.19 | (-0.54, -0.68) | 0.04 |
| | FusioN | 0.26 | (-0.38, -0.54) | 0.04 |
| ru (472) | BoN | 0.21 | (-0.50, -0.65) | 0.04 |
| | FusioN | 0.29 | (-0.31, -0.48) | 0.04 |
| zh (425) | BoN | 0.23 | (-0.45, -0.61) | 0.04 |
| | FusioN | 0.31 | (-0.28, -0.46) | 0.04 |

Table 13: Win-rates with 95% confidence intervals and standard errors for Aya-Expanse 8B vs. Gemini-2.5 Pro on mArenaHard-V2.0 ($N$=5).

## I  COMPUTE EFFICIENCY ANALYSIS

We discussed sample efficiency of FUSION relative to BON in Section 5; below we provide an analytical FLOP-based comparison showing the computational trade-offs between the two approaches.

**Framework for comparison.**   Following the scaling laws of Kaplan et al. (2020), we approximate the number of FLOPs required for a single forward pass with context length $L$ as

$$\text{FLOPs}_{\text{fw}}(L) = 2P_{\text{ne}} + 2n_\ell Ld,$$

where $P_{\text{ne}}$ denotes the number of non-embedding parameters, $n_\ell$ the number of layers, and $d$ the model dimension. To generate $T$ output tokens, we multiply $\text{FLOPs}_{\text{fw}}(L)$ by $T$.

Assuming the same model architecture and size for the fusor and the reward model, we estimate FLOPs as a function of the expected sample length and the choice of $N$. Sample length varies substantially across tasks (e.g., single sentences in machine translation versus multi-line code or longer documents in mArenaHard). We assume that FUSION does not significantly increase the final output length compared to the selected sample (as verified on mArenaHard), but produces additional tokens for evaluation or reasoning. We model this by a *reasoning factor* $r$, such that the fusor generates approximately $T = rL_s$ tokens for sample length $L_s$ (e.g., $r = 1.2$).

BON requires $N$ forward passes with inputs of length $L_s$, each producing a single output token (for simplicity; in practice a projection layer may replace the embedding layer). The total FLOPs are

$$\text{FLOPs}_{\text{BoN}} = N \cdot \text{FLOPs}_{\text{fw}}(L_s).$$

FUSION performs a single forward pass with concatenated input of length $NL_s$, and generates approximately $T = rL_s$ output tokens. The total FLOPs are therefore

$$\text{FLOPs}_{\text{Fusion}} = rL_s \cdot \text{FLOPs}_{\text{fw}}(NL_s).$$

| Lang (# pairs) | Method | Win Rate | 95% CI | Std. Err. |
|---|---|---|---|---|
| ar (479) | BoN | 0.41 | (-0.08, -0.25) | 0.04 |
| | FusioN | 0.40 | (-0.08, -0.26) | 0.04 |
| de (420) | BoN | 0.42 | (-0.06, -0.24) | 0.05 |
| | FusioN | 0.51 | (0.14, -0.05) | 0.05 |
| en (481) | BoN | 0.39 | (-0.13, -0.30) | 0.04 |
| | FusioN | 0.48 | (0.08, -0.09) | 0.04 |
| es (463) | BoN | 0.45 | (0.01, -0.17) | 0.05 |
| | FusioN | 0.53 | (0.18, -0.00) | 0.05 |
| fr (465) | BoN | 0.43 | (-0.03, -0.21) | 0.05 |
| | FusioN | 0.45 | (0.03, -0.14) | 0.05 |
| it (462) | BoN | 0.39 | (-0.11, -0.28) | 0.05 |
| | FusioN | 0.45 | (0.01, -0.17) | 0.05 |
| ja (474) | BoN | 0.41 | (-0.08, -0.25) | 0.05 |
| | FusioN | 0.42 | (-0.07, -0.24) | 0.05 |
| ko (474) | BoN | 0.43 | (-0.02, -0.19) | 0.05 |
| | FusioN | 0.45 | (-0.00, -0.18) | 0.05 |
| pt (464) | BoN | 0.39 | (-0.12, -0.30) | 0.04 |
| | FusioN | 0.44 | (-0.02, -0.20) | 0.05 |
| ru (472) | BoN | 0.39 | (-0.12, -0.29) | 0.04 |
| | FusioN | 0.43 | (-0.03, -0.21) | 0.05 |
| zh (425) | BoN | 0.47 | (0.04, -0.15) | 0.05 |
| | FusioN | 0.44 | (-0.03, -0.22) | 0.05 |

Table 14: Win-rates with 95% confidence intervals and standard errors for Command A vs. Gemini-2.5 Pro on mArenaHard-V2.0 ($N$=5).

| Model | FusioN | Skywork-V2 BoN | Internal RM BoN |
|---|---|---|---|
| Aya-Expanse 8B | **27.86** | 18.54 | 19.84 |
| Command A | **45.40** | 39.64 | 41.62 |

Table 15: Test-time scaling win-rates (%, vs. Gemini-2.5 Pro) averaged across 11 languages on mArenaHard-V2.0 with $N$=5.

Under this formulation, FUSION FLOPs grow linearly with $L_s$ through the output term and with $NL_s$ through the context length, whereas BON FLOPs scale linearly in $N$ and are independent of any additional output-length factor. As discussed in Section 6, FUSION involves one step of *sequential* scaling, while BON leverages *parallel* scaling across samples. This can be disadvantageous in compute-restricted or latency-sensitive settings. At the same time, it can be reasonable to allocate more compute to the selection process for longer inputs. This consideration also motivated our experiments on synthetic data generation, where investing additional compute in optimizing sample quality leads to improved downstream performance.

Wall-clock time and monetary cost depend on specific model implementations and hardware configurations. The above comparison therefore focuses on analytical FLOP estimates to clarify the computational trade-offs.

## J  EXTENDED RELATED WORK

**Learning from Ensembles**  The principle of learning from ensembles has led to advances in many areas of machine learning, and can be integrated into training LLMs in various forms: For example, Huang et al. (2024) fuse multiple models via their output probabilities, Lee et al. (2023) learn from a consensus of multiple teachers in self-instructing (Wang et al., 2023b), and Wan et al. (2024)

| Language | Judge | Difference (FUSION − BON) |
|----------|-------|---------------------------|
| de | Claude-4.1 | +1.4 |
|    | GPT-4o | +9.5 |
| en | Claude-4.1 | +2.9 |
|    | GPT-4o | +9.6 |
| ar | Claude-4.1 | +0.8 |
|    | GPT-4o | −1.0 |
| zh | Claude-4.1 | +1.6 |
|    | GPT-4o | −3.5 |

Table 16: Win-rate difference (FUSION − BON, %) against Gemini-2.5 Pro for Command A on mArenaHard-V2.0 under two judges.

propose a continual pretraining objective for knowledge distillation from multiple teachers. In this work, we focus on *integrative output ensembling*, where we simply provide a LLM (the fusor) the ensemble of outputs as input to integrate their strengths into a *fused* output.

**Synthesis-based ensembling**  Our approach can be seen as an instance of *Mixture-of-Agents* (MoA) (Wang et al., 2024), a framework where multiple agents organized in layers iteratively enhance the output. Our approach stands out through simplicity: We show that FUSION becomes effective already in a *single aggregation step* with a single fusor, even in diverse and challenging setups, thereby constituting an attractive alternative for BON, which is—thanks to its simplicity— a much more widely adopted framework than MoA. It is also worth noting that improvements to the sampling step in ensembling methods are directly applicable to FUSION, as they are orthogonal to the synthesis framework. These improvements target either sampling efficiency, such as pruning-based approaches proposed in Wang et al. (2025); Qiu et al. (2025), or sample quality, as explored in Minh et al. (2024); Khairi et al. (2025). We provide additional experiments on the latter in Appendix E.

*LLM-Blender* (Jiang et al., 2023) follows a similar idea, but requires two separate modules, one for pairwise ranking, and one for fusing top-ranking outputs. In contrast to our work, this framework operates on the basis of pairwise comparisons (which require training a specialized model), while we argue that the fusor should receive *all outputs at once* to best comparatively evaluate them.

Other contemporaneous related works also require training such specialized aggregator modules (Qi et al., 2025; Zhao et al., 2025; Li et al., 2025b), while our approach is effective *without any training*. These works focus primarily on verifiable tasks like math and code targeting RL or reasoning. For such specialized scenarios with expert models available, Li et al. (2025a) warn that MoA might not be sufficiently robust to lower-quality inputs. For the more diverse generative evaluation scenarios that we are targeting, however, we find that FUSION is fairly robust with respect to the teacher pool (section 5), and sampling from a single teacher—the proposed solution by Li et al. (2025a)—performs significantly worse. Jayalath et al. (2025) find that fused single-teacher rollouts can nevertheless provide valuable supervision in RL training, even without any fusor training. Overall, our work fits nicely in a stream of very recent developments discovering new possibilities of synthesis as part of the inference process. Even though the idea of FUSION is so intuitive and shared among recent works, our work advances the understanding of the inner workings and limitations of this principle. We show that implemented even in its simplest form, it brings gains in highly diverse applications for both at test-time and for driving model supervision.

**Test-time scaling**  Our approach can also be cast as combination of parallel and sequential test-time-scaling (Welleck et al., 2024; Snell et al., 2025), with $N$ parallel steps and one refinement step. Inoue et al. (2025) formulate this combination as a search problem where in a each step either more samples can be requested, or existing ones can be revisited. This poses an interesting avenue for future work, where FUSION operates with adaptive compute (rather than a fixed $N$+1) based on each input sample. This flexibility might be needed for attempts to mimic human cognitive processes more closely (Zhang et al., 2024).

**Synthetic data generation**   In the development of multilingual LLMs in particular, synthetic data generation has played a core role to reduce language disparities. For example, two recent models Apertus (Hernández-Cano et al., 2025) and EuroLLM (Martins et al., 2025), rely on EuroBlocks,[12] a collection of synthetic fine-tuning data obtained from various sources and individual teachers. Such synthetic data has also been key in improving mathematical reasoning, both monolingually (Muennighoff et al., 2025) and multilingually (Lai & Nissim, 2024; Hwang et al., 2025). Involving and ensembling multiple generations from either the same or multiple teachers in the process, as we study here, is still underexplored. For Llama 3, Grattafiori et al. (2024) report using rejection sampling (i.e. BON) for multilingual data generation. For Aya Expanse, Dang et al. (2024) report routing samples to multiple teachers (Lu et al., 2024) via multilingual BON as proposed in (Odumakinde et al., 2025), a strategy also adopted for building Tower+ (Rei et al., 2025).

## K   LLM USAGE DISCLOSURE

In this paper, we used AI in several auxiliary functions:

- Formatting of result tables in LaTeX.
- Shortening the text to fit into space limits.
- Polishing text by finding English correspondences to our non-English ideas.
- Implementation aid for the contribution analysis.
- Expansion of our initial list of related works, which we then read and carefully curated into the final related work discussion.

---

[12]https://huggingface.co/datasets/utter-project/EuroBlocks-SFT-Synthetic-1124

