# OpenReview forum: "Making, Not Taking, the Best of N"
_ICLR.cc/2026/Conference — ICLR 2026 Poster_

### Official Review · Reviewer_xyhz · 2025-11-01

**Soundness:** 4
**Presentation:** 3
**Contribution:** 3
**Rating:** 6
**Confidence:** 3

**Summary:**

This paper argues that the dominant “choose one winner” aggregation paradigm—Best-of-N or its variants like self-consistency—is wasteful and brittle because it discards potentially useful information present across candidate generations. The authors propose Fusion-of-N (FUSION): instead of selecting, a strong LLM synthesizes a final answer by integrating the best parts of N candidates. They evaluate FUSION in two common aggregation settings: (i) test-time scaling (multiple samples, one model) and (ii) synthetic data generation (multiple teachers, one student). Across 11 languages and three benchmark families (Arena-style open-ended dialogue, WMT24++ translation, and reasoning datasets like MGSM/GeoFactX), FUSION is reported to (a) improve mArena-v2 win-rates vs GEMINI2.5-PRO, (b) raise XCOMET-XL translation scores (even beating the oracle best sample on some languages), and (c) produce better synthetic datasets that yield downstream gains when used to fine-tune 7B/111B students.

**Strengths:**

### S1. Clear, compelling reformulation (selection → synthesis)
The paper articulates a neat conceptual shift: treat aggregation as synthesis rather than selection. The formalization and narrative around a polylithic view of quality (good and bad fragments within each sample) nicely motivate why a generative judge could outperform scalar scorers. The method is also simple to adopt: keep the same candidate pool, swap BON for a fusor prompt.

### S2. Broad empirical coverage with multilingual focus
The evaluation spans 11 languages, two scales for test-time scaling (8B and 111B) and multiple downstream tasks. The work puts emphasis on multilingual open-ended dialogue (mArena-Hard v2.0), WMT24++ translation (scored by XCOMET-XL), and GeoFactX/MGSM reasoning—giving the reader a more realistic view of behavior beyond English.

### S3. Strong results; synthesis can beat oracle in MT
In machine translation, FUSION beats BON by large margins and even exceeds the oracle (best single candidate given ground-truth references) in several languages (e.g., German, Russian, Chinese), establishing that generative fusion is not upper-bounded by selection. That is a crisp and important empirical takeaway.

### S4. Test-time scaling wins and consistent synthetic-data gains
On Arena-style evaluation, FUSION improves win-rates against a frontier model, Gemini 2.5-Pro, using the same N as BON; e.g., +9.5% in German for COMMAND A with N=5. Fine-tuning on FUSION-selected data yields +2.5 Arena win-rate avg and +0.8 XCOMET-XL over BON-selected data showcasing that fusion not only helps at inference, but also transfers to training.

### S5. Useful analysis of how synthesis works and when it fails
The paper inspects contribution at the character/segment level (difflib SequenceMatcher), showing the fusor often copies >80% from the presented candidates, rather than hallucinating wholecloth. Further, FUSION tends to reach BON’s quality using fewer samples. It also shows a certain size threshold—bigger fusors unlock better generative fusion—clarifying practical requirements. Moreover, the paper is candid about math being a weaker regime for out-of-the-box FUSION.

**Weaknesses:**

While I like the paper and it has strong contributions, I have a few minor suggestions. If my main requests are fulfilled (W1, W2), I would be happy to raise my score.

### W1. Dependence on a single proprietary judge.

Headline results often rely on GPT-4o for LLM-as-judge preferences (for Arena-style head-to-heads). This raises a concern: Are fusor outputs advantaged by stylistic preferences of GPT-4o (e.g. verbosity, hedging tone), rather than objective quality?
Please report, at least for a subset of languages, human eval or a second independent judge to ensure robustness (and estimate judge bias if any).

### W2. Reproducibility / openness.

A key component of the evaluation, the internal RM for scoring BoN, is proprietary or unreleased. The paper promises to release fused vs BON-selected datasets “where licenses allow,” which is good, but full reproduction remains difficult. I would strongly suggest the authors to also use an open-source reward model to make the findings reproducible for the open-source community. Please also release (i) judge prompts, (ii) fuse prompts (Table 4 already helps), (iii) a BON baseline using only public RMs, so independent researchers can validate.

### W3. Math / verifiability tasks expose a limitation.

On MGSM, BON outperforms FUSION both at test-time aggregation and after SFT. FUSION can actually degrade logical consistency by “cleaning up” reasoning steps in ways that introduce subtle arithmetic or chain-of-thought errors. This undercuts the “drop-in replacement” message: FUSION does not dominate BON in all domains, especially where correctness is binary and checkable.

I would appreciate if the authors can be more direct in the main text. For example, something like: “FUSION helps open-ended generation and translation, but BON (plus verification) is still stronger for objectively verifiable tasks such as math.”

### W4. Statistical reporting could be tighter.

Arena-style win-rates are given mostly as point estimates (+10.8, +9.5, etc.). The paper notes position biases and tie-handling, but doesn’t present full confidence intervals or absolute counts of judged pairs per language. Please add CIs or bootstrap intervals for all win-rates.




## Presentation Nitpicks:

### Typos / phrasing

1. In the fusion prompt section (Table 4), I noticed minor spelling issues such as “pormpt” → “prompt”, and awkward phrases like “scaler rater,” which should probably read “scalar rater.”
2. GeoFactX evaluation text includes a phrase like “agains the list,” which should be “against the list.”
3. MGSM discussion: “math task compared BON” should read “compared to BON.”
4. Figure captions / axis labels occasionally misspell “mArenaHard-v2” (e.g. “m-AreanHard-v2”). Please normalize.
5. “XCOMETXL” vs “XCOMET-XL” appears inconsistent; use a single canonical form in text and figures.


### Figures / presentation

1. In Figure 1 / main win-rate barplots, please annotate $N$ (e.g. N=5 samples fused), identify the judge model, and the number of judged pairwise comparisons so readers can gauge statistical strength at a glance.

2. Add error bars / confidence intervals for win-rates, or at least show per-language sample counts. Right now improvements are large, but we can’t tell which differences are stable and which are noisy.

**Questions:**

Q1 - Cost–quality trade-off: Can you plot final quality metric (win-rate vs GEMINI-2.5-PRO, XCOMET-XL, MGSM accuracy) against actual inference cost? For BON and FUSION at N = 1, 2, 5, 10? Include both FLOPs/token estimates and observed latency on your hardware. This would make the “FUSION is more sample-efficient” claim more actionable.

Q2 - Robustness to adversarial / low-quality candidates: You test robustness to weaker teacher pools, but what if a candidate is malicious: toxic, off-policy, prompt-injected, etc.? Does the fusor tend to copy problematic spans, or does it filter them out? Any ablations where you intentionally inject garbage/noisy generations into the pool?

---

> ### Author Response · Authors · 2025-11-21
> **Win-rate LLM-as-judge and open reward models (W1 & W2)**
>
> >  W1. Dependence on a single proprietary judge. Headline results often rely on GPT-4o for LLM-as-judge preferences (for Arena-style head-to-heads). This raises a concern: Are fusor outputs advantaged by stylistic preferences of GPT-4o (e.g. verbosity, hedging tone), rather than objective quality? Please report, at least for a subset of languages, human eval or a second independent judge to ensure robustness (and estimate judge bias if any).
> Thank you for raising this point. We opted to use GPT-4o as our judge for a couple of reasons: (i) it is the recommended judge by ArenaHard benchmark (ii) and it has been shown to have the highest correlation with human judges across languages in recent work [1].
>
> However there is still a possibility that the judge's biases affect the results as the reviewer pointed out. To address these we repeated the evaluation with a different judge (Claude-4.1). We find that even though the win-rates are different in magnitude, we still observe that both judges have a preference for Fusion answers compared to BoN answers.
>
> We will add the full results of the second judge to the final version of the paper. In the table below we show the win-rate in Test-time scaling of both FusioN and BoN against Gemini-2.5 Pro under both judges across a small subset  of languages (including the only two languages where BoN outperformed FusioN) and we find that FusioN outperforms BoN under Claude-4.1 for all the languages here.
>
> Finally, will also release all LLM-as-judge evaluation (from both judges) including all prompts, reasoning traces of judges and add the detailed breakdown of CIs and std-err to the paper to ensure robustness.
>
> | Language | Judge | Difference (FusioN - BoN) |
> |----------|-------|---------------------------|
> | de | Claude-4.1 | +1.4 |
> | de | GPT-4o | +9.5 |
> | en | Claude-4.1 | +2.95 |
> | en | GPT-4o | +9.6 |
> | ar | Claude-4.1 | +1.1 |
> | ar | GPT-4o | -1.0 |
> | zh | Claude-4.1 | +1.6 |
> | zh | GPT-4o | -3.5 |
>
> ---
>
> >  W2. Reproducibility / openness. A key component of the evaluation, the internal RM for scoring BoN, is proprietary or unreleased. The paper promises to release fused vs BON-selected datasets “where licenses allow,” which is good, but full reproduction remains difficult. I would strongly suggest the authors to also use an open-source reward model to make the findings reproducible for the open-source community. Please also release (i) judge prompts, (ii) fuse prompts (Table 4 already helps), (iii) a BON baseline using only public RMs, so independent researchers can validate.
> We value reproducibility of our work and we can confirm that we will release the FusioN vs BoN datasets with full details on how they were created. We also confirm that it will release both (i) judge prompts and all related details of pairwise evals including judge reasoning trace and all completions, (ii) the fusion prompts. We also want to re-state that for GeoFactX we specify all evaluation details in Appendix C using LLM-as-judge prompts from [2].
>
> Regarding the point of ‘a BON baseline using only public RMs’  we acknowledge that the use of an internal reward model may limit reproducibility, and to this end, we ran an extra post-submission experiment where we used the leading open-source reward models on RewardBench Skywork-Reward-V2-Llama-3.1-8B  for BoN. Below is the Test time scaling win-rates results using both the internal reward model and the open Skywork-V2 on ArenaHardv2, averaged across 11 languages:
>
> | Model | FusioN | Skywork-V2 BoN | Internal RM BoN |
> |-------|--------|----------------|-----------------|
> | Aya8B | 27.86 | 18.53 | 19.84 |
> | CommandA | 45.4 | 39.64 | 41.62 |
>
> ## References:
>
> [1] Kocmi, Tom, et al. "Findings of the WMT25 Multilingual Instruction Shared Task: Persistent Hurdles in Reasoning, Generation, and Evaluation." Proceedings of the Tenth Conference on Machine Translation. 2025.
>
> [2] Hwang, Jaedong, et al. "Learn globally, speak locally: Bridging the gaps in multilingual reasoning." arXiv preprint arXiv:2507.05418 (2025).

---

> ### Author Response · Authors · 2025-11-21
> **MGSM results and statistical reporting (W3 & W4)**
>
> >  W3. Math / verifiability tasks expose a limitation. On MGSM, BON outperforms FUSION both at test-time aggregation and after SFT. FUSION can actually degrade logical consistency by “cleaning up” reasoning steps in ways that introduce subtle arithmetic or chain-of-thought errors. This undercuts the “drop-in replacement” message: FUSION does not dominate BON in all domains, especially where correctness is binary and checkable.I would appreciate if the authors can be more direct in the main text. For example, something like: “FUSION helps open-ended generation and translation, but BON (plus verification) is still stronger for objectively verifiable tasks such as math.”
>
> The reviewer raises an interesting point regarding FusioN performance on MGSM with ‘degrade logical consistency’.  Our finding however suggests that this may not be the case based on the following reasons:
> - On GeoFactX, which measures both the reasoning quality and final answer correctness and is much more challenging and less saturated of a benchmark, FusioN wins over BoN both in reasoning quality and in final answer score. This confirms that FusioN does not hurt the reasoning consistency and is also applicable to verifiable and checkable benchmarks.
>
> - Additionally, for the majority of the MGSM experiments (Fig 8 and Fig 9), FusioN does not strictly underperform BoN, but rather performs on par, given that standard errors from bootstrap resampling (Fig 8) are quite large on this small benchmark (250 prompts).
>
> We also thank the reviewer for the suggestion on the writing, and while we discuss these points in Appendix E, will take these advice to tighten and clarify our reporting on the MGSM benchmark and we will add the recently released MMATH benchmark [1] as a more challenging and reliable benchmark for the final version.
>
> ---
>
> >  W4. Statistical reporting could be tighter. Arena-style win-rates are given mostly as point estimates (+10.8, +9.5, etc.). The paper notes position biases and tie-handling, but doesn’t present full confidence intervals or absolute counts of judged pairs per language. Please add CIs or bootstrap intervals for all win-rates.
>
> We agree with the reviewer 's point on the importance of confidence intervals , particularly in win-rates evaluation. We include significance testing for WMT using comet-compare and add std-err bars to both our MGSM and ArenaHardv2 results across the work (e.g Table 1, Figure 5, Figure 8, etc..) .  For further robustness we can include the calculated 95% confidence interval (following the recommendation by [2]) and the standard error for all our win-rate evaluations. We will add the full table to the final version of the paper. Below is an example of these values for 4 languages in the Test time scaling setup.
>
> | Lang (# pairs) | Method | win  | 95% CI          | std-err |
> |----------------|--------|------|------------------|---------|
> | en (481)       | BoN    | 0.39 | (-0.13, -0.3)    | 0.04    |
> | en (481)       | FusioN | 0.48 | (0.08, -0.09)    | 0.04    |
> | ar (479)       | BoN    | 0.41 | (-0.08, -0.25)   | 0.04    |
> | ar (479)       | FusioN | 0.40 | (-0.08, -0.26)   | 0.04    |
> | zh (425)       | BoN    | 0.47 | (0.04, -0.15)    | 0.05    |
> | zh (425)       | FusioN | 0.44 | (-0.03, -0.22)   | 0.05    |
> | ja (474)       | BoN    | 0.41 | (-0.08, -0.25)   | 0.05    |
> | ja (474)       | FusioN | 0.42 | (-0.07, -0.24)   | 0.05    |
>
> We would also like to point out that CI reported above are based on the comparison of the respective method with Gemini-2.5-pro and are not a direct comparison between BoN and FusioN. Direct comparison CIs however will be added to Figure 14 and 15 in the appendix where we are directly comparing BoN vs FusioN.
>
> ## References:
>
> [1] Luo, Wenyang, et al. "MMATH: A Multilingual Benchmark for Mathematical Reasoning." arXiv preprint arXiv:2505.19126 (2025).
>
> [2] Miller, Evan. "Adding error bars to evals: A statistical approach to language model evaluations." arXiv preprint arXiv:2411.00640 (2024).

---

> ### Author Response · Authors · 2025-11-21
> **Answers to reviewer xyhz questions**
>
> > Q1 - Cost–quality trade-off: Can you plot final quality metric (win-rate vs GEMINI-2.5-PRO, XCOMET-XL, MGSM accuracy) against actual inference cost? For BON and FUSION at N = 1, 2, 5, 10? Include both FLOPs/token estimates and observed latency on your hardware. This would make the “FUSION is more sample-efficient” claim more actionable.
>
> Plotting quality metrics against actual inference cost becomes difficult given our diverse experimental setup. We provide a detailed FLOP efficiency analysis in our response to reviewer jD1s.  The TLDR is that we will tighten our wording around efficiency to clarify that in terms of FLOPs per N samples, BoN is more efficient, but in terms of benchmark gains per N samples, FusioN wins up to around N=10.
>
>
> ---
>
> > Q2 - Robustness to adversarial / low-quality candidates: You test robustness to weaker teacher pools, but what if a candidate is malicious: toxic, off-policy, prompt-injected, etc.? Does the fusor tend to copy problematic spans, or does it filter them out? Any ablations where you intentionally inject garbage/noisy generations into the pool?
>
> The FusioN prompt explicitly instructs the fusor to discard problematic content, making robustness to toxic candidates dependent on the fusor's instruction-following ability. In early small-scale experiments, we found that our fusor choice (Command-A) is robust to toxic and off-policy injected candidates.
>
> ---
>
> > Typos / phrasing Figures / presentation
>
> Thank you for spotting these typos and the suggestions for better presentation  will consider  them for the final version.

---

> ### Comment · Reviewer_xyhz · 2025-11-23
>
> Thank you for running the additional experiments and for addressing each of my questions. Almost all my concerns have been resolved and I will update my score.
>
> In the confidence interval reported, the values (for example, (-0.13, -0.3) in the first row) is unclear to me. Is it from negative 0.13 to negative 0.3? Does that mean their model loses to Gemini-2.5-Pro most of the time? I am quite confused.
>
> While the authors say that "CI reported above are based on the comparison of the respective method with Gemini-2.5-pro and are not a direct comparison between BoN and FusioN.", it is kind of confusing. I would appreciate if the authors can share the direct CIs here, which they have promised to add in the Appendix.
>
> I agree with the author's clarification on MGSM where the error bars are indeed large and the benchmark is almost saturated (I  didn't notice this in my initial review). I look forward to the results on MMATH instead.

---

> > ### Author Response · Authors · 2025-11-24
> >
> > We thank the reviewer for the positive feedback and updating the score.
> >
> > Regarding the confidence interval reported, the reviewer's interpretation is correct. The table reported above shows win-rates and CIs for FusioN/BoN against `Gemini-2.5-pro`, which is a stronger model than `Command-A`, which we use for FusioN/BoN. Hence, the win-rates are usually `< 0.5` and the CI is in the *negative*. We can see from the table and Figure 1 in the paper that test-time scaling with FusioN gets `Command-A` performance close to `Gemini-2.5-pro` across the majority of languages.
> >
> > However, as the reviewer pointed out, the CIs from directly comparing FusioN against BoN will be a clearer way to present this. The table below shows this comparison on mArenaHard-v2 test-time scaling setup for Command-A (see Figure 14) across all the languages. We see here the expected *positive* CIs and win-rates that are `> 0.5`. We will add this table to the paper for both models.
> >
> > | Lang | win  | 95% CI | std_err |
> > |------|------|----    |---------|
> > | en   | **0.51**| (0.17, 0.0) | 0.043 |
> > | de   | 0.50| (0.16, -0.01) | 0.043 |
> > | fr   | **0.52**| (0.18, 0.02) | 0.043 |
> > | es   | 0.48| (0.13, -0.03) | 0.043 |
> > | ru   | **0.51**| (0.18, 0.02) | 0.043 |
> > | ja   | 0.49| (0.14, -0.03) | 0.043 |
> > | zh   | **0.55**| (0.25, 0.09) | 0.043 |
> > | ar   | 0.48| (0.15, -0.01) | 0.042 |
> > | pt   | **0.52**| (0.21, 0.05) | 0.042 |
> > | it   | **0.55**| (0.24, 0.07) | 0.043 |
> > | ko   | **0.52**| (0.2, 0.03) | 0.043 |

---

### Official Review · Reviewer_digP · 2025-11-01

**Soundness:** 3
**Presentation:** 4
**Contribution:** 3
**Rating:** 8
**Confidence:** 4

**Summary:**

This paper argues that the dominant Best‑of‑N (BON) framing of generate N candidates and select one throws away useful signal. The authors propose Fusion‑of‑N (FUSION): a training‑free, prompt‑driven synthesis step in which a fusor LLM reads the prompt and the N candidates and writes a new answer that integrates the best parts and discards the worst.

They evaluate FUSION in two settings that currently rely on BON: (i) test‑time scaling (sampling multiple candidates from the same model and aggregating), and (ii) synthetic data generation (sampling across a pool of teacher models and aggregating for SFT). Experiments span 11 languages, three task families (open‑ended generation on mArenaHard‑v2, WMT24++ translation, and factual/math reasoning), and two model scales (8B and 111B).

FUSION consistently beats BON at the same sampling budget and, on WMT, sometimes exceeds the “oracle” single candidate scored against references, establishing that selection is not an upper bound. The paper further analyzes when FUSION works best (larger fusors, small N, diverse teacher pools) and where it struggles (close‑ended math)

**Strengths:**

- With a single prompt, FUSION is drop‑in for selection and consistently improves win‑rates on open‑ended prompts and XCOMET‑XL scores on translation across 11 languages. The fact that FUSION sometimes surpasses the reference‑scored oracle in WMT (German/Russian/Chinese; Fig. 2, p. 5) is a sharp, convincing result showing synthesis > selection.
- The paper evaluates both test‑time and data‑generation use cases, the two places BON is used most heavily today, and does so across 11 languages

**Weaknesses:**

- The attribution method (difflib) cannot detect paraphrastic reuse or semantic fusion. Also, the fusion prompt is English‑only, and Table 3 (p. 8) notes slightly lower “language correctness,” suggesting the prompt could disadvantage lower‑resource languages. A multilingual fusion prompt and a semantic contribution analysis (e.g., span‑level alignment or entailment) would strengthen claims about “polylithic” quality.
- The core sales point is sample efficiency at small N, yet the paper does not report wall‑clock or token‑level costs for: sampling N candidates, concatenating them into a long‑context window, and running fusion. Since BON can parallelize N samples and fusion adds a sequential long‑context pass, end‑to‑end time/compute curves vs. quality are necessary to substantiate the “efficient” claim (authors briefly note this qualitatively, and Fig. 5 shows win‑rates only).
- The authors observe a mild position bias (Fig. 11b, p. 25) and randomize candidate order, but do not explore robust mitigations (e.g., segmented fusion with random reshuffles, majority‑vote over multiple shuffled fusions)

**Questions:**

- Beyond the position‑bias probe, did you (a) repeat Arena comparisons with two heterogeneous judges and report agreement, or (b) conduct a small‑scale human‑blind study to validate GPT‑4o outcomes, as recommended by recent LLM‑as‑Judge surveys? Please add such a calibration or discuss why it’s not feasible.
- Can you provide token‑level accounting for (i) sampling N candidates, (ii) concatenation length, and (iii) fusion generation—and compare end‑to‑end latency vs. BON at equal quality? This matters because FUSION’s long‑context pass is not trivially parallelizable, and likely adds significant user compute time
- Table 3 notes slightly lower “language correctness” for FUSION, and the fusion prompt is English‑only (Table 4). Have you tried localized fusion prompts (per language) or a multilingual safety/style rubric? If so, do GeoFactX non‑English results improve?
- What happens if 1–2 of 5 teachers inject subtly wrong facts or unsafe content? Does FUSION filter them or amplify them, and can adding a pre‑filter materially help?
- Could you complement string‑matching with semantic span alignment (e.g., NLI‑based attribution or claim‑level voting) to better characterize “fusion” vs. “copy”? This will clarify whether FUSION is truly synthesizing or mostly selecting + rephrasing

---

> ### Author Response · Authors · 2025-11-21
> **Answers to reviewer digP weaknesses and questions**
>
> > [...] the paper does not report wall‑clock or token‑level costs for: sampling N candidates, concatenating them into a long‑context window, and running fusion. Since BON can parallelize N samples and fusion adds a sequential long‑context pass [...]
>
> We thank the reviewer for raising this important point about compute efficiency. We provide a comprehensive analytical FLOP comparison in our response to reviewer jD1s, where we show that FusioN FLOPs grow linearly with sample length while BoN FLOPs remain constant. As noted there, FusioN uses sequential scaling (requiring one long-context pass) while BoN leverages parallel scaling, which can be advantageous in time-sensitive applications. We acknowledge that our claim of "efficiency" requires clarification: while BoN is more efficient in terms of FLOPs per N samples, FusioN achieves superior benchmark gains per N samples. We will revise the paper to emphasize this distinction more clearly.
>
> > The authors observe a mild position bias (Fig. 11b, p. 25) and randomize candidate order, but do not explore robust mitigations (e.g., segmented fusion with random reshuffles, majority‑vote over multiple shuffled fusions)
>
> We quantify position bias in Figure 11b and observe that samples in positions 1-2 have higher contribution (22–23%) compared to positions 3-5 (11–14%). To ensure this does not affect our results, we shuffle the input order to the fusor in every experiment, effectively mitigating position bias across all reported results. While more sophisticated approaches could be used, our current shuffling strategy already ensures that position effects do not systematically advantage any particular sample ordering.
>
> > Beyond the position‑bias probe, did you (a) repeat Arena comparisons with two heterogeneous judges and report agreement, or (b) conduct a small‑scale human‑blind study to validate GPT‑4o outcomes, as recommended by recent LLM‑as‑Judge surveys [...]
>
> We address this concern by repeating the Arena evaluations with Claude-4.1 as a second judge, as detailed in our response to reviewer jD1s. The results show consistent signals across both judges: FusioN outperforms BoN in the majority of languages, though the magnitudes vary depending on the judge. This cross-judge validation demonstrates that our findings are not artifacts of GPT-4o-specific biases.
>
> > Can you provide token‑level accounting for (i) sampling N candidates, (ii) concatenation length, and (iii) fusion generation—and compare end‑to‑end latency vs. BON at equal quality?[...]
>
> We agree with reviewer points here. We provide a detailed analytical FLOP comparison in our response to reviewer jD1s, showing that FusioN FLOPs grow linearly with sample length while BoN FLOPs remain constant. FusioN uses sequential scaling (requiring one long-context pass over `N*sample_length` tokens) while BoN leverages parallel scaling, which can be advantageous in time-sensitive applications. We acknowledge that in terms of FLOPs per N samples, BoN is more efficient, but FusioN achieves superior benchmark gains per N samples.
>
>
>
> > Table 3 notes slightly lower “language correctness” for FUSION, and the fusion prompt is English‑only (Table 4). Have you tried localized fusion prompts (per language) or a multilingual safety/style rubric?[...]
>
> We thank the reviewer for this suggestion and will include an ablation on localized fusion prompts in the revised manuscript.
>
>
> > What happens if 1–2 of 5 teachers inject subtly wrong facts or unsafe content? Does FUSION filter them or amplify them, and can adding a pre‑filter materially help?
>
> The FusioN prompt (see Table 4) explicitly instructs the fusor to discard problematic content, making robustness to toxic or incorrect candidates dependent on the fusor's instruction-following ability. In early small-scale experiments, we found that our fusor choice (Command-A) is robust to unsafe content.
>
> > Could you complement string‑matching with semantic span alignment (e.g., NLI‑based attribution or claim‑level voting) to better characterize “fusion” vs. “copy”?[...]
>
> To address this point, we investigated semantic similarity between FusioN generation and each teacher generation across languages. We find that the trend is consistent, though the ranking of Qwen and Gemma sometimes changes (Gemma shows higher similarity for it, ja, ko, ar). This provides additional insight beyond surface-based analysis, which measures generally stronger contribution for DeepSeek and Qwen, with Gemma following. This could point toward more style-based similarity rather than exact wording.
>
> | Method | Command-A | DeepSeek | Qwen | Gemma | Kimi |
> |--------|-----------|----------|------|-------|------|
> | Semantic | 0.9211 | 0.8875 | 0.8580 | 0.8601 | 0.8301 |
> | Surface | 30.96 | 18.24 | 21.37 | 8.30 | 4.93 |
>
> We also investigated intra-model similarity: FusioN outputs have on average the highest similarity to all other models' outputs, while Kimi tends to be the least similar to other teachers.

---

### Official Review · Reviewer_jD1s · 2025-11-10

**Soundness:** 2
**Presentation:** 2
**Contribution:** 1
**Rating:** 2
**Confidence:** 4

**Summary:**

The paper proposes Fusion-of-N (FUSION), a paradigm that replaces the traditional Best-of-N (BON) selection strategy for LLM responses. Instead of picking a single “best” candidate among N sampled outputs, a fusor model synthesizes a new response by combining the strengths of all N candidates. The authors claim this approach better utilizes complementary information among responses.
They evaluate FUSION in two scenarios: (1) test-time scaling—sampling multiple responses from a single model and fusing them; and (2) data synthesis—fusing outputs from multiple teacher models to generate training data for a student model. Experiments span several multilingual and multi-domain tasks (open-ended generation, translation, factual QA, and math reasoning). The paper provides a universal prompt template for fusion and reports that FUSION outperforms BON on most tasks, though BON remains stronger on MGSM (math reasoning).

**Strengths:**

1. The paper identifies a real limitation of BON, the inability to exploit complementary information between candidates—and argues for synthesis rather than selection as a reasonable alternative.

2. The method is lightweight: it requires no additional training or fine-tuning, only a general “fusion prompt,” making it reproducible and accessible for practitioners.

3. The authors openly report cases where FUSION fails (e.g., MGSM), which adds transparency.

**Weaknesses:**

1. The proposed method is largely a prompt-level heuristic. The core idea of asking a strong LLM to compare and fuse multiple outputs, is conceptually simple and closely resembles prior response aggregation or mixture-of-agents paradigms. The paper mostly offers a new name and systematic evaluation rather than a substantive algorithmic innovation.

2. The BON baseline uses an unspecified reward model and selection protocol. Details such as training data, tuning strategy, and model scale are unclear, making it difficult to judge fairness. The authors only compare against small-N BON (typically N=5), omitting modern BON improvements (hierarchical sampling, de-duplication, or diverse beam reranking). All comparisons rely on a single LLM judge (GPT-4o), introducing possible evaluation bias.

3. The paper claims FUSION sometimes surpasses the oracle best candidate (highest XCOMET score among N). However, this “oracle” is defined by an automatic metric; it is not clear that human annotators would agree. This undermines one of the paper’s central claims.

4. The fusion step concatenates N long answers into a single context, increasing token length and making inference sequential and costly. The authors briefly mention sample efficiency but ignore wall-clock latency, memory, or dollar cost. In contrast, BON’s selection is trivially parallelizable.

5. FUSION underperforms BON on verifiable reasoning tasks like MGSM but outperforms on subjective or open-ended tasks. The paper fails to explain why or to propose an adaptive choice mechanism.

6. Input formatting (order of candidates, truncation, token limits, multilingual mixing) is under-specified. The authors mention minor order bias but do not quantify its effect.

**Questions:**

1. What reward model and settings were used for BON? Would a stronger public RM or combined judge close the reported gap?

2. Can you provide human or multi-reference evaluations showing that FUSION genuinely produces higher-quality translations than the best candidate?

3. Why does FUSION fail on MGSM? Is it due to logical reasoning disruption, verbosity, or hallucination?

4. Please report the average input length, latency, and cost per query for N={2,4,8}.

---

> ### Author Response · Authors · 2025-11-21
> **BoN reward model, scale and modern sampling methods**
>
> >  The BON baseline uses an unspecified reward model and selection protocol. Details such as training data, tuning strategy, and model scale are unclear, making it difficult to judge comparison fairness [...] I'm curious what reward model and settings were used for BON. Would a stronger public RM or a combined judge reduce the reported gap?
> We acknowledge the reviewer's concern about the reward model details and their effect on reproducibility. In the submitted work, we reported public benchmark scores of our internal Reward Model (RM) on RewardBench2 and mRewardBench (see footnote 1) to provide context about its capabilities.
>
> To verify that FusioN's success is not solely due to our choice of RM in BoN, we conducted a post-submission experiment using the currently leading open-source reward model on RewardBench2: `Skywork-Reward-V2-Llama-3.1-8B`.
>
> We report test-time scaling win-rate results (N=5) for both the internal reward model and the open Skywork-V2 on mArenaHardv2, averaged across 11 languages:
>
> | Model | FusioN | Skywork-V2 BoN | Internal RM BoN |
> |-------|--------|----------------|-----------------|
> | Aya8B | 27.86 | 18.53 | 19.84 |
> | CommandA | 45.4 | 39.64 | 41.62 |
>
> These results confirm FusioN's superiority compared to BoN under SOTA reward models and enhance reproducibility.
>
> ---
>
> >  The authors only compare against small-N BON (typically N=5), omitting modern BON improvements (hierarchical sampling, de-duplication, or diverse beam reranking)
> ### using small N=5:
> We are mainly  concerned with studying use cases that have practical relevance and potential for adoption in deployed systems. While we study the effect of scaling up to N=20 in Figure 5 in the paper, we find diminishing returns of test-time scaling, hence we focus on the range of N where added compute has the largest impact.
>
> ### Modern BoN method:
> We agree with the reviewer that the simple temperature-sampling Reward Model BoN can be modified to incorporate more complex search methods, such as those proposed by [1]  and [2]. We will include these, and any others the reviewer may suggest, in our discussion of related work. However, we would like to point out two key limitations in comparing these methods to FusioN:
>
> These search-focused techniques are not applicable to the multi-teacher setup, which is key to FusioN's downstream benefits (Table 2; see also [3]).
> These modern improvements have not been tested and applied in the broader setup across languages and tasks, limiting their comparability to FusioN in our experiments.
>
> More generally, we expect all improvements to the sampling strategy for BoN to transfer to FusioN as well. To demonstrate this, we implement two modifications: Hedged Sampling [4] and Min-p [5], both shown to improve sample quality robustly across languages and tasks.
>
> We report test-time scaling results (win-rates over greedy outputs) on mArenaHardv2 averaged across 7 languages (en,de,fr,es,ru,zh,ja). We observe that better sampling techniques not only benefit BoN performance, but also boost FusioN. In fact, FusioN leverages the higher-quality samples even more effectively than BoN:
> | Model | BoN | FusioN |
> |-------|-----|--------|
> | Aya8b | +12.5 | +33.0 |
> | CommandA | +5.0 | +16.6 |
>
>
> ## References:
>
> [1] Wang, Yiming, et al. "Sampling-efficient test-time scaling: Self-estimating the best-of-n sampling in early decoding." arXiv preprint arXiv:2503.01422 (2025).
>
> [2] Qiu, Jiahao, et al. "Treebon: Enhancing inference-time alignment with speculative tree-search and best-of-n sampling." arXiv preprint arXiv:2410.16033 (2024).
>
> [3] Odumakinde, Ayomide, et al. "Multilingual arbitration: Optimizing data pools to accelerate multilingual progress." Proceedings of the 63rd Annual Meeting of the Association for Computational Linguistics (Volume 1: Long Papers). 2025.
>
> [4] Khairi, Ammar, et al. "When Life Gives You Samples: The Benefits of Scaling up Inference Compute for Multilingual LLMs." Proceedings of the 2025 Conference on Empirical Methods in Natural Language Processing. 2025.
>
> [5] Nguyen, Minh Nhat, et al. "Turning up the heat: Min-p sampling for creative and coherent llm outputs." arXiv preprint arXiv:2407.01082 (2024).

---

> ### Author Response · Authors · 2025-11-21
> **Win-rates and WMT evaluations robustness**
>
> >  All comparisons rely on a single LLM judge (GPT-4o), introducing possible evaluation bias
> We thank the reviewer for pointing out potential biases in LLM-as-judge evaluation. We agree that any single judge is likely biased. GPT-4o has been shown to have the highest correlation with human judges across languages in recent work [1], making it a reasonable choice for the mArenaHard benchmark.
>
> To address this concern, we repeated the evaluations with another judge (Claude-4.1). The table below shows the win-rate differences between FusioN and BoN in the test-time scaling setup where we compare against Gemini-2.5 Pro. The signal is fairly consistent across both judges: FusioN outperforms BoN in the majority of languages, though the magnitudes vary depending on the judge.
>
> | Language | Judge | Difference (FusioN - BoN) |
> |----------|-------|---------------------------|
> | de | Claude-4.1 | +1.4 |
> | de | GPT-4o | +9.5 |
> | en | Claude-4.1 | +2.95 |
> | en | GPT-4o | +9.6 |
> | ar | Claude-4.1 | +1.1 |
> | ar | GPT-4o | -1.0 |
> | zh | Claude-4.1 | +1.6 |
> | zh | GPT-4o | -3.5 |
>
> ---
>
> >  The paper claims FUSION sometimes surpasses the oracle best candidate (highest XCOMET score among N). However, this “oracle” is defined by an automatic metric; it is not clear that human annotators would agree. This undermines one of the paper’s central claims [...] Can you provide human or multi-reference evaluations showing that FUSION truly produces higher-quality translations than the best candidate?
> Regarding the relevance for humans, we can estimate it with the help of the MT-Thresholds tool [2], e.g. a system-level +0.5 XCometXL (the case for German (de), see Fig 2) difference is estimated to have 65% accuracy with humans, or a +0.8 XCometXL difference (ja) is estimated to have 74% accuracy with humans.
>
> We compute BLEU as a complementary MT metric (as implemented in sacrebleu, with tokenizers configured according to target language) for the test-time scaling experiments with Command A reported in Figure 2 to confirm that FusioN can outperform the oracle translation (as chosen by the respective metric). We find that for 4/10 languages, FusioN outperforms oracle (i.e. the sampled translation that scores highest under sentence-level BLEU) in aggregated scores:
>
>
> | Language | Oracle | BoN   | FusioN        |
> |----------|--------|-------|---------------|
> | de       | 29.39  | 24.98 | **29.77** (26%) |
> | es       | 41.41  | 36.65 | 39.06 (12%)    |
> | fr       | 38.00  | 33.00 | 35.63 (12%)    |
> | it       | 34.86  | 28.43 | 34.20 (17%)    |
> | ja       | 22.61  | 17.35 | **22.80** (34%) |
> | ko       | 24.61  | 18.54 | **25.41** (36%) |
> | pt       | 27.94  | 24.01 | 26.92 (14%)    |
> | ru       | 21.13  | 16.96 | 21.09 (28%)    |
> | zh       | 36.92  | 26.59 | **37.68** (34%) |
>
>
> We added the % of sentences where FusioN outperforms oracle on the test set in brackets, showing that in all languages, FusioN is able to outperform the oracle in 12% or more cases. We would like to emphasize that it does not mean that FusioN must always outperform the oracle, but that it is a possibility due to its design (see Section 2), while BoN can never do so.
>
> ## References:
>
> [1] Kocmi, Tom, et al. "Findings of the WMT25 Multilingual Instruction Shared Task: Persistent Hurdles in Reasoning, Generation, and Evaluation." Proceedings of the Tenth Conference on Machine Translation. 2025.
>
> [2] Kocmi, Tom, et al. "Navigating the metrics maze: Reconciling score magnitudes and accuracies." arXiv preprint arXiv:2401.06760 (2024).

---

> ### Author Response · Authors · 2025-11-21
> **FusioN Efficiency Claims**
>
> > The fusion step concatenates N long answers into a single context, increasing token length and making inference sequential and costly. The authors briefly mention sample efficiency but ignore wall-clock latency, memory, or dollar cost. In contrast, BON’s selection is trivially parallelizable. I think this point needs to be clarified.
> We agree with the reviewer that FusioN incurs higher compute costs for the selection step. While a wall-clock or dollar cost comparison might be too dependent on individual model and compute environment choices, we conduct an analytical FLOP comparison.
>
> ### Framework for comparison
>
> Following the scaling laws by [1], we assume that the number of FLOPs for a single forward pass can be calculated as:
>
> ```
> FLOPs_fw = 2*non_embed_params + 2*n_layer*context_length*model_dim
> ```
>
> To predict T tokens, we multiply `FLOPs_fw * T`. If we assume the same model architecture and size for the fusor and reward model, we can estimate FLOPs depending on expected sample length and our choice of N.
>
> Sample length varies widely across tasks (e.g., single sentences for machine translation versus multi-line code snippets or documents for Arena). We assume that FusioN does not significantly increase output length compared to the samples it receives (verified on Arena), but does spend additional tokens for evaluation/reasoning, proportional to the sample length (e.g., `reas_factor = 1.2`).
>
> ### Comparing BoN and FusioN
>
> BoN requires N forward passes with `sample_length`-long inputs, yielding T=1 output token each (this is a simplification; in practice we have a projection layer replacing the embedding layer):
>   ```
>   FLOPs_BoN = N * FLOPs_fw(sample_length)
>   ```
>
> FusioN requires 1 forward pass with an `N*sample_length` input, yielding approximately `T = reas_factor * sample_length` output tokens:
>   ```
>   FLOPs_FusioN = reas_factor * sample_length * FLOPs_fw(N*sample_length)
>   ```
>
> This means that the FusioN FLOPs grow linearly with growing sample_length, while BoN FLOPs remain constant. As pointed out in Related Work, FusioN has one step of *sequential* scaling, while BoN only leverages *parallel* scaling. While this can be disadvantageous in compute-restricted environments and time-sensitive applications, it is intuitive to spend more compute on the process of finding the best sample for longer inputs. This also motivated our experiments with synthetic data generation, where it pays off investing more compute into optimizing data quality, as it leads to significant downstream gains.
>
> ### Revisions to the paper
>
> We will tighten the wording in our paper regarding efficiency and include this analysis in the appendix. In the paper we talk about “sample efficiency”, i.e. under the same sampling budget (N, not total compute budget) it outperforms BoN, but that is not controlling for total compute. We will emphasize more that FusioN incurs a higher cost.
>
>
> ## References:
> [1] Kaplan, Jared, et al. "Scaling laws for neural language models." arXiv preprint arXiv:2001.08361 (2020).

---

> ### Author Response · Authors · 2025-11-21
> **FusioN on MGSM**
>
> >  FUSION underperforms BON on verifiable reasoning tasks like MGSM but outperforms on subjective or open-ended tasks. The paper fails to explain why or to propose an adaptive choice mechanism [...] Why does FUSION fail on MGSM? Is it due to logical reasoning disruption, verbosity, or hallucination? More in-depth analysis and discussion are expected to be provided here.
> We respectfully disagree, this statement is not accurate: MGSM is only one of two verifiable reasoning tasks investigated in this work. On GeoFactX, which tests factual reasoning and is much more challenging and less saturated of a benchmark, FusioN wins over BoN both in reasoning quality and in final answer score. Furthermore, we have tested on WMT which is neither open-ended nor subjective, as it is evaluated with respect to a reference translation.
>
> For the majority of the MGSM experiments (Fig 8 and Fig 9), FusioN does not strictly underperform BoN, but rather performs on par, given that standard errors from bootstrap resampling (Fig 8) are quite large on this small benchmark (250 prompts). We understand that in the face of the large, consistent and robust gains on other benchmarks, this is easy to understand as a failure case. We suspect it is rather due to the benchmark than the FusioN method. For finetuning, we discuss this already in Appendix G. MGSM is a saturated and small benchmark with little statistical power (e.g. the 0.4% difference in MGSM Accuracy for de in Fig 8 between FusioN and BoN is caused by a single example that FusioN gets wrong and for zh, the language with the biggest gap, it is 7 examples) - we admit that the y-axis scale is misleading). Therefore, the conclusion “FusioN underperforms BON on verifiable reasoning tasks” is not reliably supported by our data.
>
> We will add the recently released MMATH benchmark [1] as a more challenging and reliable benchmark for the final version.
>
> Regardless of the reliability of MGSM as a benchmark, we were curious about the ‘adaptive choice mechanism” that the reviewer brought up. This led us to experiment with a combined approach that filters the inputs to fusion by the reward score (from the N=5 pool, we filter out any sample with a reward score lower than an adaptive threshold ,5%, calculated based on the highest score in the pool ). We found that in 5 out of 7 tested languages (with no effect en and de) the pre-filtering brought minor improvements for Command-A, see table below. We will be able to verify on MMATH if this points to the fusor model in isolation being subpar on judging mathematical correctness, and whether FusioN would benefit from adapting the FusioN prompt (Table 4) specifically to math (e.g. “The fused generation should be mathematically correct.”).
>
> | language_code | method                       | value  |
> |---------------|------------------------------|--------|
> | fr            | FusioN                       | 85.20  |
> | fr            | FusioN_Adaptive_reward_score | 86.80  |
> | es            | FusioN                       | 90.80  |
> | es            | FusioN_Adaptive_reward_score | 92.00  |
> | ru            | FusioN                       | 94.00  |
> | ru            | FusioN_Adaptive_reward_score | 94.40  |
> | ja            | FusioN                       | 90.00  |
> | ja            | FusioN_Adaptive_reward_score | 90.40  |
> | zh            | FusioN                       | 90.00  |
> | zh            | FusioN_Adaptive_reward_score | 90.40  |
>
>
> ## References:
> [1] Luo, Wenyang, et al. "MMATH: A Multilingual Benchmark for Mathematical Reasoning." arXiv preprint arXiv:2505.19126 (2025).

---

> ### Author Response · Authors · 2025-11-21
> **Answers to reviewer jD1s questions**
>
> > Input formatting (order of candidates, truncation, token limits, multilingual mixing) is under-specified. The authors mention minor order bias but do not quantify its effect.
>
> Figure 11b quantifies the position bias as measured with surface-level contribution statistics. Samples in position 1 and 2 tend to have higher contribution (22–23%), while those in 3-5th position have a lower contribution (11–14%). What this analysis does not take into account is the similarity between samples and the diminishing novelty of contents: It could be an efficient and valid approach to start with the first samples as a draft and refine according to differences in later samples (compare Figures 12 and 13 for an example). To make sure this is not affecting the results, we have shuffled the input order to the fusor in every experiment.
>
> ---
>
> > If convenient, please report the average input length, latency, and cost per query for N={2,4,8}. More detailed implementation and experiment details should be provided, especially considering the code is not provided in the supplementary materials.
>
>
> We describe our experimental setup in Appendix A , B C and D and we will release the prompts, completions and judge evaluations. More specific details such as average length or latency per query will be harder to estimate considering our diverse setup of tasks, languages and model sizes.
>
> ---
>
> > What's the difference between this works and the line of works on parallel thinking? This part of analysis is expected in the revised manuscript.
>
> The ParaThinker work [1] introduces a model-native architecture that generates multiple reasoning paths in one forward pass and then fuses them, while our approach differs in that we do not train a dedicated fusion-capable model. Instead, our method leverages multiple teacher models and aggregates their outputs externally. This distinction means we can reuse off-the-shelf models without requiring the structural modifications  that ParaThinker employs. We thank the reviewer for bringing this work to our attention and we will add it to our revised manuscript’s related work.
>
>
> ## References:
> [1] Wen, Hao, et al. "Parathinker: Native parallel thinking as a new paradigm to scale llm test-time compute." arXiv preprint arXiv:2509.04475 (2025)

---

### Author Response · Authors · 2025-11-21
**Summary of our response**

We thank the reviewers  for the appreciation and the constructive feedback for our work.

Reviewers found our synthesis-over-selection paradigm “compelling” as a “lightweight” and “accessible” **“drop-in” for the established Best-of-N paradigm**. They expressed that they appreciated the **“broad empirical coverage” and transparency of our experiments** with the proposed FusioN method spanning tasks from multiple domains and languages for multiple model scales, in the “two places BON is used most heavily today”, test-time scaling and synthetic data generation. They found the results “strong”, “consistent”, and “convincing”, and our analysis “useful”, “showing synthesis > selection”.

The primary concerns were about the **robustness of our empirical evaluations**, such as the dependence on particular evaluation choices e.g. LLM judge, reward model or sampling strategy. We are happy to report that these suggestions were helpful in solidifying our findings, and highlighting the benefits of FusioN even more:

- We added results for the *public SOTA reward model* [Reviewers jD1s and xyzh], which confirms the superiority of FusioN compared to BoN and enhances reproducibility.

- We added results for an *improved sampling strategy* [Reviewer jD1s] showing that FusioN benefits from sampling advances more than BoN.

- We added results with a *second public and competitive LLM judge* [Reviewers jD1s and xyzh], that also prefers FusioN over BoN.

- We added machine translation evaluations with a *different metric* [Reviewer jD1s], under which FusioN is also able to outperform the oracle.

Details can be found within the individual responses. We also discuss the outcomes on MGSM, correct misunderstandings regarding performance on reasoning tasks, report confidence intervals, and clarify the compute requirements.

We hope that the reviewers will find the answers they were looking for, and would be grateful for a continued discussion.

---

### Author Response · Authors · 2025-11-29
**Summary of Discussion -  MMATH and prompt tuning results**

We thank the reviewers for their valuable feedback and **Reviewer xyhz** for the productive discussion and **updated score** (*before the leak*). We encourage the AC to take into account that they updated their score from 6 to 8, as we have addressed their concerns. **Reviewer j1Ds** (who gave an initial 2) had not had a chance to react to our rebuttal, but we suspect we addressed their concerns too as they were mostly identical with reviewer xyhz’s concerns.

As a follow-up from the discussion with reviewer `xyhz` and reviewer `j1Ds`, we provide additional results on the **MMATH benchmark**, showing that FusioN does outperform BoN in math reasoning on more difficult tasks (one of the core concerns of all reviewers), along with an **ablation on fusion prompt design** (as requested by reviewer `digP`) as we indicated in our initial responses. We will include both in the final manuscript.

---

## MMATH Overall Results

We compared FusioN and BoN performance on the MMATH benchmark across **10 languages** with generations from four teacher models (Command-A, Qwen-3, Gemma-3, Deepseek-V3). **FusioN achieves +1.0% higher accuracy than BoN** on average (std-err: 0.6%). The Oracle outperforms both methods by approximately +6%, indicating both methods have substantial room for improvement in this benchmark.
|  Method   |   Accuracy |   std-err |
|:---------|-----------:|----------:|
|  BoN      |     84.045 |     0.602 |
|  **FusioN**   |     **85.127** |     0.585 |
|  *Oracle*   |     *90.562* |     0.481 |

Overall, this shows that there is *no inherent limitation of FusioN for math reasoning*.

We do find that the gap to BoN is consistently smaller across MGSM and MMATH than on generative tasks, which is because it aligns less with the motivation of polylithic quality: It’s more likely that when the first reasoning steps are already bad, that the model would later produce a bad reasoning trace or answer - while in generative tasks outside reasoning, the quality of the entire output is more compositional, the first sentence can be stylistically poor, while the following sentence might be just fine. Therefore, *FusioN has more potential* to improve over BoN in such generative tasks.

---

## Fusion Prompt Ablation

We performed an ablation on prompt design, testing two modifications: (1) adapting the prompt wording for math problems (emphasizing reasoning accuracy over style), and (2) translating the prompt into the target language. The table below shows results for test-time scaling with Command-A; neither modification produces significant improvements with both changes still within the standard error of the original prompt score. This shows that performance of FusioN is not sensitive to the details in prompt formulation, which is attractive in practice because it removes the burden of prompt tuning.

| Method                                     |   Accuracy   |   std-err |
|:-------------------------------------------|-------------:|----------:|
| FusioN                                     |       72.772 |     2.302 |
| FusioN + Math Prompt                       |       72.906 |     2.297 |
| FusioN + Math Prompt + Target Language     |       72.772 |     2.298 |

---

### Meta-Review · Area_Chair_hZ8p · 2026-01-07

**Summary:**

This paper presents a prompting technique to combine the best of n generations from a language model for better test time scaling as well as synthetic data generation. Experiments on tasks such as open-ended generation, translation, factual QA demonstrate the effectiveness of fusion-of-n over best-of-n, though math reasoning is better for best-of-n. Many of the experiments are done in multilingual settings, including machine translation.

**Reviewer Concerns:**

**Addressed**
1. Reviewer jD1s’s concerns about details in the method by varia sampling methods, number of samples, reward model (also LLM judge?) etc.
2. Results on other benchmarks, based on xyhz’s comments.
3. Additional analysis, based on digP’s comments about semantic vs surface level combinations.


**Unaddressed**
1. All three reviewers point out that since the method relies on a prompt-level heuristic, there might be additional noise introduced at this step. To the best of my understanding the author response does not address this beyond a very cursory explanation. For xyhz, the authors do provide a secondary fusion prompting model, but there is no method outlined on how to take into account two different models for fusion prompting, i.e. only two prompting methods are compared, but the method is not robust to the use of any single alternative.
2. The method seems expensive, but no time efficiency results are reported. The author response outlines an analysis in terms of flops, but no wall-clock times are reported. This is particularly important to know how much time goes into creating a good “fusion” of responses.
3. Most importantly, as pointed out by jd1s, this method seems to not be algorithmically novel, though the empirical investigations are pretty thorough. Most glaringly, the authors seemed to have missed out on several related papers which present (variants of) this prompting method as a baseline:
    - Saha, S., Levy, O., Celikyilmaz, A., Bansal, M., Weston, J., & Li, X. (2024, June). Branch-solve-merge improves large language model evaluation and generation. In Proceedings of the 2024 Conference of the North American Chapter of the Association for Computational Linguistics: Human Language Technologies (Volume 1: Long Papers) (pp. 8352-8370).
    - Xie, Yuxi, Kenji Kawaguchi, Yiran Zhao, James Xu Zhao, Min-Yen Kan, Junxian He, and Michael Xie. "Self-evaluation guided beam search for reasoning." Advances in Neural Information Processing Systems 36 (2023): 41618-41650.
    - Ghosh, S., Warraich, S. S., Tarsadiya, D., Yauney, G., & Swayamdipta, S. (2025). Sample, Align, Synthesize: Graph-Based Response Synthesis with ConGrs. arXiv preprint arXiv:2510.03527.
    - Thirukovalluru, R., Huang, Y., & Dhingra, B. (2024). Atomic self-consistency for better long form generations. arXiv preprint arXiv:2405.13131.

**Reviewer Scores:**

Xyhz: This reviewer explicitly reports interest in increasing their score.

Digp: The authors present a lot of new results to respond to this reviewer. Some major concerns of this reviewer are not adequately addressed, however: wall-time reporting, need for a human-blind study, “language correctness” for FUSION, teachers inject subtly wrong facts or unsafe content. I’m not sure this reviewer would have increased their score.

jd1s: However, the method itself is not novel and not much has been done to argue for the heuristic nature of the method, which is ultimately dependent on prompting another language model to fuse a given language model’s responses. In that regard, I cannot help but agree with the reviewer digP, and I’m not sure they would have increased their score to above acceptance.

---

### Decision · Program_Chairs · 2026-01-26

Accept (Poster)